# In-Motion Initial Alignment Method Based on Multi-Source Information Fusion for Special Vehicles

**DOI:** 10.3390/e27030237

**Published:** 2025-02-25

**Authors:** Zhenjun Chang, Zhili Zhang, Zhaofa Zhou, Xinyu Li, Shiwen Hao, Huadong Sun

**Affiliations:** College of Missile Engineering, Xi’an Research Institute of High Technology, Xi’an 710025, China; changzj2105@163.com (Z.C.); 13772426956@163.com (Z.Z.); zzftxy@163.com (Z.Z.); wenjy70796@163.com (S.H.); sunhuadong0927@outlook.com (H.S.)

**Keywords:** strapdown inertial navigation system, in-motion alignment, multi-source information fusion, federal Kalman filter, fault diagnosis and isolation

## Abstract

To address the urgent demand for autonomous rapid initial alignment of vehicular inertial navigation systems in complex battlefield environments, this study overcomes the technical limitations of traditional stationary base alignment methods by proposing a robust moving-base autonomous alignment approach based on multi-source information fusion. First, a federal Kalman filter-based multi-sensor fusion architecture is established to effectively integrate odometer, laser Doppler velocimeter, and SINS data, resolving the challenge of autonomous navigation parameter calculation under GNSS-denied conditions. Second, a dual-mode fault diagnosis and isolation mechanism is developed to enable rapid identification of sensor failures and system reconfiguration. Finally, an environmentally adaptive dynamic alignment strategy is proposed, which intelligently selects optimal alignment modes by real-time evaluation of motion characteristics and environmental disturbances, significantly enhancing system adaptability in complex operational scenarios. The experimental results show that the method proposed in this paper can effectively improve the accuracy of vehicle-mounted alignment in motion, achieve accurate identification, effective isolation, and reconstruction of random incidental faults, and improve the adaptability and robustness of the system. This research provides an innovative solution for the rapid deployment of special-purpose vehicles in GNSS-denied environments, while its fault-tolerant mechanisms and adaptive strategies offer critical insights for engineering applications of next-generation intelligent navigation systems.

## 1. Introduction

Navigation is a technology to obtain the motion state of moving objects and guide the route, which plays an important role in human exploration of the unknown world and social development [1]. Inertial navigation technology has been widely employed in the military industry and civilian sectors due to its high stability, dependability, autonomy, and all-weather capability, among other advantages. From submarines, ships, aircraft, and missiles to medical equipment, drones, cell phones, etc., there is inertial navigation equipment that exists and even a great show [2]. An inertial navigation system must be initially aligned before the carrier enters navigation, which can provide the initial attitude for the carrier navigation on the one hand and provide the initial orientation for other navigation equipment at the end of the travel on the other hand. The essence is to determine the attitude between the target and the navigation reference system and provide the initial value for the subsequent navigation and positioning [3].

There are various ways to categorize initial alignment [4,5]. According to the SINS working conditions, it can be divided into static pedestal, shaking pedestal, and in-motion alignment. The static pedestal alignment relies on the external environment, where the shaking vibration interference is very small and is generally applicable to the indoor initial alignment. The shaking pedestal refers to the carrier being in the stopping and shaking state due to the impact of the engine vibration, the personnel getting on and off the carrier, and the airflow, and it is the most commonly used in the current SINS initial alignment in the practice [6]. In-motion alignment refers to the carrier being in the traveling state and in the complex external environment to obtain the carrier attitude. Especially for special vehicles, the use of in-motion alignment technology can shorten the preparation time to improve the system’s survivability and rapid mobility, and it is a research hotspot to be solved by the vehicle-mounted SINS initial alignment [7].

The vector attitude determination alignment method does not need the initial information of attitude, and it is an effective means of coarse alignment for vehicle-mounted SINS. For alignment in motion, although the alignment in the inertial system can isolate the effect of base sway, it needs vehicle-mounted velocimetry equipment to obtain the velocity of the carrier motion and compensate for the interference of the carrier motion because the accelerometer output contains the information of the carrier motion. Generally, the vehicle-mounted velocimetry equipment includes GNSS, odometer (OD), and Laser Doppler Velocimeter (LDV), in which GNSS can provide the carrier’s position and velocity in the geographic coordinate system, and it is not affected by the scale factor and installation error. Many scholars have carried out research on the GNSS-assisted initial alignment in motion algorithms due to its high accuracy [8,9,10,11]. However, it is not applicable to the alignment in motion for special vehicles because of the autonomous ability and invisibility. Compared with GNSS, the OD and LDV do not need external information and can realize autonomous alignment. References [12,13,14,15] studied odometer-assisted initial alignment in motion for vehicle-mounted SINS. LDV-assisted initial alignment in motion on submerged ships has been studied in references [16,17,18], and references [19,20] introduced the LDV into vehicle-mounted SINS initial alignment to solve the problem of unstable odometer scale factor. Although the vectors attitude determination does not need to be implemented at small misalignment angles, the error parameters of the inertial devices and velocimetry equipment are still dependent on the prior calibration and cannot be estimated in real time during the alignment. Compared with the vector attitude determination in motion, the optimal estimation method can expand the error parameters of the inertial devices and speed measurement equipment to state vectors, which can be estimated during the alignment process, thus realizing the function of fault diagnosis and isolation. The purpose of the initial alignment is to obtain the attitude of the carrier relative to the navigation coordinate system, which usually requires the velocity provided by the velocimetry equipment as the measurement information to be filtered and estimated. Commonly used auxiliary velocimetry devices are the same as vector attitude determination alignment in motion, and references [21,22,23] have studied the optimal estimation of alignment in motion assisted by GNSS, OD, and LDV, respectively. Reference [24] considered the impact of the complex traveling environment on the filtering model and used the strong tracking filtering algorithm to realize the alignment in motion. Since the optimal estimation method of Kalman filtering depends on the linear error model, it is only applicable to fine alignment. Considering the fault tolerance of the system, if some auxiliary devices fail, the federal Kalman filter (FKF) can be applied to reconfigure the system, and if all auxiliary devices are unable to provide auxiliary information, the carrier-constrained Kalman filtering can still be completed by using the measurement information that the vertical and lateral actual velocity of the carrier is basically zero during the traveling process [25,26]. Electronic map-assisted alignment in motion requires the prior collection of a digital map within the usage area, which limits its application [27,28]. In addition, visual sensors are also available aids for alignment in motion but are still in the theoretical research stage [29,30,31].

However, longer alignment times are often required to achieve higher accuracy in practical use, and enhancing the robustness of the alignment system to improve fault tolerance comes with the risk of increasing the computational cost or even sacrificing some of the accuracy. Therefore, it is crucial to select different alignment strategies according to the actual operation condition of the SINS. In view of this, this paper proposes an initial alignment method based on multi-source information fusion for special vehicles in motion, designs a system fault-tolerant scheme through federal Kalman filtering, and gives a strategy for adaptively selecting the alignment method under the complex and multifarious actual operation conditions, which improves the alignment accuracy while enhancing the adaptability and robustness of the SINS.

The remainder of the paper is organized as follows. Section 2 discusses in detail the fundamentals and theoretical basis of initial alignment. The proposed initial alignment method in motion based on multi-source information fusion, the system fault-tolerant design, and the adaptive alignment strategy are described in Section 3. Section 4 carries out the vehicle-mounted initial alignment experiments, which verifies the feasibility and validity of the methodology proposed in this paper, and the conclusions are formed in Section 5, which summarizes the whole paper.

## 2. Initial Alignment Method of SINS

Initial alignment for SISN is actually a process of finding the reference navigation coordinate system and determining the spatial orientation of the body coordinate system with respect to the navigation coordinate system. Prior to the establishment of the basic model of the vehicle-mounted SINS, a clear definition of the relevant coordinate system is provided [32].

Inertial coordinate system (*i*): The geocentric inertial coordinate system is chosen, the origin *O_i_* is located in the center of the Earth, *O_i_x_i_* points to the equinox in the equatorial plane, *O_i_z_i_* points to the north of the Earth’s axis, and *O_i_y_i_* forms a right-handed orthogonal coordinate system with the first two. Neglecting the motion of the Earth’s center and the change in the direction of the Earth’s axis, the *i* system is immobile.Earth coordinate system (*e*): The Earth-centered Earth fixed system is chosen, the origin *O_e_* is located at the center of the Earth, *O_e_x_e_* is the line of intersection between the equatorial plane and the starting meridian plane, *O_e_z_e_* points northward on the Earth’s axis, and *O_e_y_e_* and the two previous ones form a right-handed orthogonal coordinate system.Geographic coordinate system (*g*): The origin *O_g_* is located in the mass center of the carrier, the *O_g_x_g_* axis is located in the horizontal plane pointing to the east, the *O_g_y_g_* axis is located in the horizontal plane pointing to the north, and the *O_g_z_g_* axis forms a right-handed orthogonal coordinate system with the first two, i.e., the “East-North-Upon (*E-N-U*)” coordinate system, which is completely coincident with the navigation coordinate system (*n*).Body coordinate system (*b*): The mass center of the carrier is chosen as the origin *O_b_*, the *O_b_x_b_* axis points to the right, the *O_b_x_b_* axis points to the front along the longitudinal axis of the carrier, and the *O_b_z_b_* axis forms a right-handed orthogonal coordinate system with the first two, i.e., the “Right-Forward-Upon (*R-F-U*)” coordinate system.IMU coordinate system (*s*): The origin is defined as the mass center of IMU, and the three axes are the directions of the nominal sensitivity axes of the three accelerometers. Since the IMU is mounted on the vehicle, there is a mounting error angle between the *s* system and the *b* system, but the *b* system can be substituted for the *s* system after strict calibration compensation.Earth coordinate system at the initial moment (*e*_0_): The origin *O_e_*_0_ is located at the center of the Earth, the *O_e_*_0_*x_e_*_0_ axis is located in the equatorial plane pointing to the local meridian plane aligned with the initial moment, the *O_e_*_0_*z_e_*_0_ axis is pointing to the north of the Earth’s axis, and the *O_e_*_0_*y_e_*_0_ axis forms a right-handed orthogonal coordinate system together with the two previous ones.Navigation coordinate system at the initial moment (*n*_0_): The *n* system that is aligned to the initial moment and is thereafter immobile with respect to the Earth’s surface.

### 2.1. Fundamentals of SINS Initial Alignment

Even in the parking state of practical applications, due to engine vibration, personnel getting on and off the vehicle, gusts of wind, and other factors, the initial alignment of the vehicle-mounted SINS faces disturbances such as angular wobbling and line vibration, which seriously affects the accuracy of the initial alignment. In order to overcome the problem of a wobbling base, the gravity vectors at two different moments in the inertial system (both of them are not parallel in the *i* system) are selected to complete the dual-vector attitude determination. For this purpose, two new coordinate systems need to be defined: the inertial coordinate system at the initial moment (*i*_0_ system, which coincides with the *e*_0_ system at the initial moment of alignment, after which the three axes are solidified to the inertial space and remain unchanged) and the body coordinate system at the initial moment (*i_b_*_0_ system, which solidifies the *b* system to the inertial space at the initial moment of alignment and remains unchanged). With the help of the *i*_0_ and *i_b_*_0_ systems, the attitude matrix $C_{b}^{n}$ can be decomposed as follows:(1)$$C_{b}^{n}=C_{i_{0}}^{n}C_{i_{b0}}^{i_{0}}C_{b}^{i_{b0}}$$
where $C_{x}^{y}$ denotes the transformation matrix from the *x* system to the *y* system, *x* and *y* are the coordinate systems already defined in the paper, and the same for the latter. The following three matrices are solved separately to complete the initial alignment of SINS.

According to the chain multiplication rule for matrices, $C_{i_{0}}^{n}$ satisfies the following [33]:(2)$$C_{i_{0}}^{n}=C_{e}^{n}C_{n_{0}}^{e}C_{e_{0}}^{n_{0}}C_{i_{0}}^{e_{0}}$$

Let the geographic latitude of the initial alignment starting moment be $L_{0}$, the longitude be $\lambda_{0}$, the geographic latitude of the t-moment be L, and the longitude be λ. The specific form of matrices $C_{e}^{n}$, $C_{n_{0}}^{e}$, $C_{e_{0}}^{n_{0}}$ and $C_{i_{0}}^{e_{0}}$ can be seen from reference [34].

Due to the initial alignment of the wobble base in the parked state and the constant geographic location of the carrier, we can yield the following:(3)$${\hat{C}}_{i_{0}}^{n}=\left[\begin{array}{ccc} -sin(\omega_{ie}t) & cos(\omega_{ie}t) & 0\\ -sinLcos(\omega_{ie}t) & -sinLsin(\omega_{ie}t) & cosL\\ cosLcos(\omega_{ie}t) & cosLsin(\omega_{ie}t) & sinL \end{array}\right]$$
where $\omega_{ie}$ is the angular velocity of the Earth’s rotation.

The matrix $C_{b}^{i_{b0}}$ on the right side of the equality sign of Equation (1) satisfies the following:(4)$${\dot{C}}_{b}^{i_{b0}}=C_{b}^{i_{b0}}\left(\omega_{ib}^{b}\times \right)$$
where ${\hat{C}}_{b}^{i_{b0}}$ can be solved in real time according to the gyroscope output ${\tilde{\omega }}_{ib}^{b}$ and the initial value of $C_{b}^{i_{b0}}\left(t_{0}\right)=I$ based on the attitude update algorithm.

The matrix $C_{i_{b0}}^{i_{0}}$ on the right side of the equality sign of Equation (1) is a constant matrix, which is solved as follows.

Simultaneous differentiation of both sides of the $v^{n}=C_{i_{b0}}^{n}v^{i_{b0}}$ yields the following:(5)$${\dot{v}}^{n}=C_{i_{b0}}^{n}({\dot{v}}^{i_{b0}}+\omega_{ni_{b0}}^{i_{b0}}\times v^{i_{b0}})$$
where $v^{n}$ and $v^{i_{b0}}$ are the projection of the velocity vector in the *n* system and the *i_b_*_0_ system, respectively, $\omega_{ni_{b0}}^{i_{b0}}$ is the rotation angular rate of the *i_b_*_0_ system relative to the *n* system, and $\omega_{ni_{b0}}^{i_{b0}}=-\omega_{in}^{i_{b0}}$. Substitute Equation (5) into the specific force equation of the Strapdown inertial navigation [35] and multiply both sides by $C_{n}^{i_{b0}}$, then there is the following:(6)$${\dot{v}}^{i_{b0}}+(C_{i_{0}}^{i_{b0}}\omega_{ie}^{i_{0}})\times v^{i_{b0}}-C_{b}^{i_{b0}}f^{b}=C_{i_{0}}^{i_{b0}}C_{n}^{i_{0}}g^{n}$$
where ***f****^b^* is the specific force output of the accelerometer and ***g****^n^* is the gravity acceleration vector in the *n* system. From Equation (6), it can be seen that the left side of the equation is the projection of the gravity vector in the *i_b_*_0_ system, which is the result of removing the interference of the carrier motion from the accelerometer output, so the following definition is made:(7)$$\left\{\begin{array}{l} g^{i_{b0}}={\dot{v}}^{i_{b0}}+(C_{i_{0}}^{i_{b0}}\omega_{ie}^{i_{0}})\times v^{i_{b0}}-C_{b}^{i_{b0}}f^{b}\\ g^{i_{0}}=C_{n}^{i_{0}}g^{n} \end{array}\right.$$

Then Equation (6) can be converted into the following:(8)$$g^{i_{0}}=C_{i_{b0}}^{i_{0}}g^{i_{b0}}$$
where $g^{i_{0}}$, $g^{i_{b0}}$ are the gravity acceleration vectors under the *i*_0_ and *i_b_*_0_ systems, respectively. Since the gravity vector moves along the cone in inertial space as the Earth rotates at the nonpolar points, $g_{1}^{i_{0}}$, $g_{2}^{i_{0}}$, $g_{1}^{i_{b0}}$, $g_{2}^{i_{b0}}$ are selected in the *i*_0_ and *i_b_*_0_ systems, respectively, which are not parallel to each other at different moments, and Equation (9) can be obtained by dual-vector attitude determination.(9)$${\hat{C}}_{i_{b0}}^{i_{0}}=\left[\begin{array}{ccc} \frac{g_{1}^{i_{0}}}{\left\|g_{1}^{i_{0}}\right\|} & \frac{g_{1}^{i_{0}}\times g_{2}^{i_{0}}}{\left\|g_{1}^{i_{0}}\times g_{2}^{i_{0}}\right\|} & \frac{g_{1}^{i_{0}}\times g_{2}^{i_{0}}\times g_{1}^{i_{0}}}{\left\|g_{1}^{i_{0}}\times g_{2}^{i_{0}}\times g_{1}^{i_{0}}\right\|} \end{array}\right]{\left[\begin{array}{ccc} \frac{{\tilde{g}}_{1}^{i_{b0}}}{\left\|{\tilde{g}}_{1}^{i_{b0}}\right\|} & \frac{{\tilde{g}}_{1}^{i_{b0}}\times {\tilde{g}}_{2}^{i_{b0}}}{\left\|{\tilde{g}}_{1}^{i_{b0}}\times {\tilde{g}}_{2}^{i_{b0}}\right\|} & \frac{{\tilde{g}}_{1}^{i_{b0}}\times {\tilde{g}}_{2}^{i_{b0}}\times {\tilde{g}}_{1}^{i_{b0}}}{\left\|{\tilde{g}}_{1}^{i_{b0}}\times {\tilde{g}}_{2}^{i_{b0}}\times {\tilde{g}}_{1}^{i_{b0}}\right\|} \end{array}\right]}^{-1}$$

${\hat{C}}_{i_{b0}}^{i_{0}}$, ${\hat{C}}_{i_{0}}^{n}$ and ${\hat{C}}_{b}^{i_{b0}}$ derived from Equations (3) and (4) are substituted into Equation (1) to obtain the attitude matrix ${\hat{C}}_{b}^{n}$, which completes the initial alignment of dual-vector attitude determination.

### 2.2. Initial Alignment in Motion of Inertial Systems

Compared with the initial alignment of SINS in the traditional parking state, the gravitational acceleration cannot be obtained directly in motion due to the influence of the carrier’s motion acceleration, and external velocimetry equipment is required to assist the carrier in completing the alignment in motion. Unlike the shaking base, there is motion interference during alignment on the moving base, and external velocimetry compensation is needed when ${\hat{g}}^{i_{b0}}(t)$ calculated by Equation (7). Since the GNSS assistance relies on satellite information and it is impossible to realize the alignment in motion or even obtain the incorrect alignment results when the satellite signals are disturbed or spoofed, this paper selects the OD and LDV with strong autonomy to assist the initial alignment.

${\hat{g}}^{i_{b0}}(t)$ is the projection of the gravity vector in the *i_b_*_0_ system in Equation (7), which is obtained by accelerometer measurements after removing the harmful component of the carrier motion when alignment in motion. The velocity $v^{b}$ in the *b* system is obtained by direct measurement of the OD/LDV, and the velocity $v^{i_{b0}}$ in the *i_b_*_0_ system in the harmful acceleration satisfies the following:(10)$$v^{i_{b0}}={\hat{C}}_{b}^{i_{b0}}v^{b}$$
where ${\hat{C}}_{b}^{i_{b0}}$ is obtained from Equation (4). Substituting Equation (10) into Equation (7) yields ${\hat{g}}^{i_{b0}}(t)$, which in turn yields ${\hat{C}}_{i_{b0}}^{i_{0}}$ from the dual-vector or multi-vector attitude determination, while considering ${\hat{C}}_{i_{0}}^{n}$ and ${\hat{C}}_{b}^{i_{b0}}$ from Equations (3) and (4), and substituting into Equation (1) yields the pose matrix ${\hat{C}}_{b}^{n}$.

In the initial alignment in motion of vehicle-mounted SINS, in addition to the velocity information of the carrier provided by the velocimetry equipment and thus compensating for the harmful acceleration, it is also necessary to consider the carrier displacement. The alignment algorithm in Equation (1) utilizes the inertial solidification idea to decompose $C_{b}^{n}$ into three matrix multiplication forms, in which the $C_{i_{0}}^{n}$ is related to the real-time position of the carrier in motion, which is updated by Equation (3), and if the change of carrier displacement in the process of the initial alignment is ignored, it will bring the initial alignment solving error. If the initial alignment in motion is completed by GNSS assistance, the geographic position information can be obtained directly, while OD/LDV can only provide the velocity of the carrier in the *b* system, and cannot provide the geographic position, so it is also necessary to obtain the geographic position by attitude transformation and integral iteration for OD/LDV-assisted alignment in motion [36].

### 2.3. Optimal Estimation of Initial Alignment

Although the inertial system alignment does not need initial attitude information and can complete the alignment in motion, it is mostly used in the coarse alignment due to the limited accuracy and the fact that the system relies on prior calibration and cannot monitor the parameter status in real time. The optimal estimation method of initial alignment mostly adopts the indirect method, i.e., the misalignment angle is estimated in real time with the assistance of velocimetry equipment based on the SINS model and parameter statistical characteristics and corrected in the process of navigation updating, which is capable of obtaining higher accuracy and estimating all the state coefficients in real time, with the function of fault monitoring.

#### 2.3.1. Kalman Filter Model of Initial Alignment

The initial alignment of the wobble base in the parking state can ignore the SINS position error and the carrier speed $v^{n}=0$, then the following relationship is obtained based on the basic principles of strapdown inertial navigation.(11)$$\left\{\begin{array}{l} {\dot{\phi }}^{n}=-\omega_{ie}^{n}\times \phi^{n}+M_{dav}\delta v^{n}-C_{b}^{n}C_{s}^{b}\varepsilon^{s}\\ \delta {\dot{v}}^{n}=f^{n}\times \phi^{n}-2\omega_{ie}^{n}\times \delta v^{n}+C_{b}^{n}C_{s}^{b}\nabla^{s} \end{array}\right.$$
where $\phi^{n}$ is the attitude misalignment angle, $\omega_{ie}^{n}$ is the angular velocity of the Earth’s rotation, $\delta v^{n}$ is the velocity error, $f^{n}$ is the specific force output of the accelerometer, $\nabla^{s}$ is the zero bias of the accelerometer, $\varepsilon^{s}$ is the gyroscope drift error, and ***M****_dav_* is the matrix reflecting the effect of the velocity error on the misalignment angle, the specific form of which can be seen from reference [34].

The 12-dimensional state vector $X={\left[\begin{array}{cccc} {\left(\phi^{n}\right)}^{T} & {\left(\delta v^{n}\right)}^{T} & {\left(\varepsilon^{s}\right)}^{T} & {\left(\nabla^{s}\right)}^{T} \end{array}\right]}^{T}$ is selected and the state equation is constructed as follows:(12)$$\dot{X}=AX+Bw$$
where $A=\left[\begin{array}{cccc} -\omega_{ie}^{n}\times & M_{dav} & -C_{b}^{n}C_{s}^{b} & 0_{3\times 3}\\ f^{n}\times & -2\omega_{ie}^{n}\times & 0_{3\times 3} & C_{b}^{n}C_{s}^{b}\\ 0_{3\times 3} & 0_{3\times 3} & 0_{3\times 3} & 0_{3\times 3}\\ 0_{3\times 3} & 0_{3\times 3} & 0_{3\times 3} & 0_{3\times 3} \end{array}\right]$ is state transition matrix, $B=\left[\begin{array}{cccc} -C_{b}^{n}C_{s}^{b} & 0_{3\times 3} & 0_{3\times 3} & 0_{3\times 3}\\ 0_{3\times 3} & C_{b}^{n}C_{s}^{b} & 0_{3\times 3} & 0_{3\times 3}\\ 0_{3\times 3} & 0_{3\times 3} & 0_{3\times 3} & 0_{3\times 3}\\ 0_{3\times 3} & 0_{3\times 3} & 0_{3\times 3} & 0_{3\times 3} \end{array}\right]$ is noise matrix, $w={\left[\begin{array}{cccc} {\left(w_{g}^{s}\right)}^{T} & {\left(w_{a}^{s}\right)}^{T} & 0_{1\times 3} & 0_{1\times 3} \end{array}\right]}^{T}$ is system noise, $w_{g}^{s}$ and $w_{a}^{s}$ are random noise errors for accelerometers and gyroscopes, respectively.

In the parking state, the vehicle velocity $v^{n}$ is zero, and if the SINS solved speed ${\hat{v}}^{n}$ is not zero at this time, the equation ${\hat{v}}^{n}-v^{n}=\delta v^{n}$ reflects the influence of SINS inertial device error and solving error, which can be used as a measurement ***Z***, and construct the measurement equation as follows.(13)$$Z={\hat{v}}^{n}=HX+u$$
where $H=\left[\begin{array}{cccc} 0_{3\times 3} & I_{3\times 3} & 0_{3\times 3} & 0_{3\times 3} \end{array}\right]$ is the measurement matrix, and ***u*** is measurement noise.

#### 2.3.2. Optimal Estimation of Initial Alignment in Motion

The carrier velocity is not zero, and the filtering model needs to be reconstructed for alignment in motion. The velocity error under the *n* system is first modeled, and the odometer velocity model and error model are as follows [37]:(14)$${\hat{v}}_{O_{s}}^{s}=v_{O_{s}}^{s}+\delta K_{b}C_{b}^{s}v^{b}+C_{b}^{s}\left(\delta \alpha \times \right)v^{b}-\omega_{eb}^{s}\times \delta L^{s}-\varepsilon^{s}\times L^{s}$$(15)$$\delta {\hat{v}}_{O_{s}}^{s}=\delta K_{b}C_{b}^{s}v^{b}+C_{b}^{s}\left(\delta \alpha \times \right)v^{b}-\omega_{eb}^{s}\times \delta L^{s}-\varepsilon^{s}\times L^{s}$$
where ${\hat{v}}_{O_{s}}^{s}$ is the actual velocity output of the OD, $v_{O_{s}}^{s}$ is the measurement value of the OD at *O_s_* point under *s* system, $\delta K_{b}$ is the scale factor error, δα is the calibration error, i.e., the installation error angle of the corresponding *b* system of the velocimetry equipment relative to the *s* system, δL is the rod-arm error, $\omega_{eb}^{s}$ is the angular speed of rotation of the *b* system relative to the *e* system, and the rest of the symbols are the same as the meaning above.

Transforming the OD velocimetry model of Equation (14) to the *n* system yields the following:(16)$${\hat{v}}_{O_{s}}^{n}={\hat{C}}_{s}^{n}{\hat{v}}_{O_{s}}^{s}=C_{s}^{n}\left(I+\phi^{n}\times \right)\left(v_{O_{s}}^{s}+\delta v_{O_{s}}^{s}\right)$$

Substituting Equation (15) into Equation (16), it can be approximated that(17)$${\hat{v}}_{O_{s}}^{n}\approx v_{O_{s}}^{n}+v_{O_{s}}^{n}\times \phi^{n}+\delta K_{b}C_{b}^{n}v^{b}+C_{s}^{n}\left(L^{s}\times \right)\varepsilon^{s}-C_{b}^{n}\left(v^{b}\times \right)\delta \alpha -C_{s}^{n}\left(\omega_{eb}^{s}\times \right)\delta L^{s}$$

The velocity error under the *n* system can be obtained as(18)$$\begin{array}{c} \delta v_{O_{s}}^{n}=v_{O_{s}}^{n}\times \phi^{n}+\delta K_{b}C_{b}^{n}v^{b}+C_{s}^{n}\left(L^{s}\times \right)\varepsilon^{s}-C_{b}^{n}\left(v^{b}\times \right)\delta \alpha -C_{s}^{n}\left(\omega_{eb}^{s}\times \right)\delta L^{s}\\ =v_{O_{s}}^{n}\times \phi^{n}+C_{b}^{n}\left[v^{b}\delta K_{b}-\left(v^{b}\times \right)\delta \alpha \right]+C_{s}^{n}\left(L^{s}\times \right)\varepsilon^{s}-C_{s}^{n}\left(\omega_{eb}^{s}\times \right)\delta L^{s} \end{array}$$

Considering that $v^{b}={\left[\begin{array}{ccc} 0 & v & 0 \end{array}\right]}^{T}$, there is the following equation in Equation (18).(19)$$v^{b}\delta K_{b}-\left(v^{b}\right)\delta \alpha =\left[\begin{array}{c} -v\delta \alpha_{z}\\ v\delta K_{OD}\\ v\delta \alpha_{x} \end{array}\right]=\left[\begin{array}{ccc} 0 & 0 & -v\\ v & 0 & 0\\ 0 & v & 0 \end{array}\right]\left[\begin{array}{c} \delta K_{b}\\ \delta \alpha_{x}\\ \delta \alpha_{z} \end{array}\right]=M_{v}\left[\begin{array}{c} \delta K_{b}\\ \delta \alpha_{x}\\ \delta \alpha_{z} \end{array}\right]$$
where ***M****_v_* denotes the matrix composed of forward velocity v, as shown in Equation (19), it can be seen that in the mounting error of IMU and carrier, only $\delta \alpha_{x}$ and $\delta \alpha_{z}$ have an effect on the velocity error, corresponding to pitch and yaw, respectively, and the rolling mounting error angle has no effect on the velocity error. Thus, a new error vector $\delta K_{\alpha }={\left[\begin{array}{ccc} \delta K_{b} & \delta \alpha_{x} & \delta \alpha_{z} \end{array}\right]}^{T}$ can be constructed by combining $\delta K_{b}$, $\delta \alpha_{x}$, and $\delta \alpha_{z}$, and expanding the position error $\delta p^{n}$ and the rod-arm error $\delta L^{s}$ together to the estimated state vector, which can be obtained as follows:(20)$$X_{1}={\left[\begin{array}{ccccccc} {\left(\phi^{n}\right)}^{T} & {\left(\delta v^{n}\right)}^{T} & {\left(\delta p^{n}\right)}^{T} & {\left(\varepsilon^{s}\right)}^{T} & {\left(\nabla^{s}\right)}^{T} & {\left(\delta K_{\alpha }\right)}^{T} & {\left(\delta L^{s}\right)}^{T} \end{array}\right]}^{T}$$

Substituting Equation (19) into Equation (17) and Equation (18), the velocity and velocity error under the *n* system can be obtained as(21)$${\hat{v}}_{O_{s}}^{n}\approx v_{O_{s}}^{n}+v_{O_{s}}^{n}\times \phi^{n}+C_{b}^{n}M_{v}\delta K_{\alpha }+C_{s}^{n}\left(L^{s}\times \right)\varepsilon^{s}-C_{s}^{n}\left(\omega_{eb}^{s}\times \right)\delta L^{s}$$(22)$$\delta v_{O_{s}}^{n}=v_{O_{s}}^{n}\times \phi^{n}+C_{b}^{n}M_{v}\delta K_{\alpha }+C_{s}^{n}\left(L^{s}\times \right)\varepsilon^{s}-C_{s}^{n}\left(\omega_{eb}^{s}\times \right)\delta L^{s}$$

The constructed measurement ***Z***_1_ is(23)$$Z_{1}={\hat{v}}^{n}-{\hat{v}}_{O_{s}}^{n}$$

When there is no error in the velocimetry device, there is $v^{n}=v_{O_{s}}^{n}$, and Equation (23) can be transformed to(24)$$Z_{1}=\left(v^{n}+\delta v^{n}\right)-\left(v_{O_{s}}^{n}+\delta v_{O_{s}}^{n}\right)=\delta v^{n}-\delta v_{O_{s}}^{n}=H_{1}X_{1}+u_{1}$$
where $H_{1}=\left[\begin{array}{cccc} -v^{n}\times & I_{3\times 3} & 0_{3\times 3} & \begin{array}{cccc} -C_{s}^{n}\left(L^{s}\times \right) & 0_{3\times 3} & -C_{b}^{n}M_{v} & C_{s}^{n}\left(\omega_{eb}^{s}\times \right) \end{array} \end{array}\right]$.

Let the scale factor and installation error $\delta K_{\alpha }$ and the rod-arm error $\delta L^{s}$ be constant values, then the error equation of SINS and velocimetry equipment is(25)$$\left\{\begin{array}{ccc} {\dot{\phi }}^{n} & = & -\omega_{in}^{n}\times \phi^{n}+M_{dav}\delta v^{n}+M_{dap}\delta p^{n}-C_{s}^{n}\varepsilon^{s}-C_{s}^{n}w_{g}^{s}\\ \delta {\dot{v}}^{n} & = & f^{n}\times \phi^{n}+\left[v^{n}\times M_{dav}-2\omega_{ie}^{n}\times -\omega_{en}^{n}\times \right]\delta v^{n}+v^{n}\times M_{dvp}\delta p^{n}+C_{s}^{n}\nabla^{s}+C_{s}^{n}w_{a}^{s}\\ \delta {\dot{p}}^{n} & = & M_{dpv}\delta v^{n}+M_{dpp}\delta p^{n}\\ {\dot{\varepsilon }}^{s} & = & 0\\ \nabla^{s} & = & 0\\ \delta {\dot{K}}_{\alpha } & = & 0\\ \delta {\dot{L}}^{s} & = & 0 \end{array}\right.$$
where $\omega_{in}^{n}$ represents the rotation angular velocity of *n* system relative to *i* system, which contains two parts, one is the rotation angular velocity $\omega_{ie}^{n}$ of the *n* system caused by the rotation of the Earth, and the rotation angular velocity $\omega_{en}^{n}$ of the *n* system caused by the bending of the Earth’s surface as the SINS moves near the Earth’s surface, where ***M****_dap_* reflects the effect of the positional error on the alignment angle, ***M****_d__vp_* reflects the effect of the positional error on the velocity error, ***M****_dp__v_* reflects the effect of the velocity on the positional error, and ***M****_d__pp_* reflects the relationship between the positional error and its own differential. Please refer to reference [34] for their specific forms.

The state equation is constructed as(26)$${\dot{X}}_{1}=A_{1}X_{1}+B_{1}w_{1}$$
where $A_{1}=\left[\begin{array}{cccccc} -\omega_{in}^{n}\times & M_{dav} & M_{dap} & -C_{s}^{n} & 0_{3\times 3} & 0_{3\times 6}\\ f^{n}\times & v^{n}\times M_{dav}-2\omega_{ie}^{n}\times -\omega_{en}^{n}\times & v^{n}\times M_{dvp} & 0_{3\times 3} & C_{s}^{n} & 0_{3\times 6}\\ 0_{3\times 3} & M_{dpv} & M_{dpp} & 0_{3\times 3} & 0_{3\times 3} & 0_{3\times 6}\\ 0_{12\times 3} & 0_{12\times 3} & 0_{12\times 3} & 0_{12\times 3} & 0_{12\times 3} & 0_{12\times 6} \end{array}\right]$, $B_{1}=\left[\begin{array}{ccc} -C_{s}^{n} & 0_{3\times 3} & 0_{3\times 15}\\ 0_{3\times 3} & C_{s}^{n} & 0_{3\times 15}\\ 0_{15\times 3} & 0_{15\times 3} & 0_{15\times 15} \end{array}\right]$, $w_{1}={\left[\begin{array}{ccc} {\left(w_{g}^{s}\right)}^{T} & {\left(w_{a}^{s}\right)}^{T} & 0_{1\times 15} \end{array}\right]}^{T}$. The Kalman filtering model for vehicle-mounted SINS alignment in motion can be constructed based on Equations (24) and (26). Since the IMU drifts more and more with the increase of measurement time, but its accuracy is higher in a short time, while the accuracy of the velocimetry device does not diverge with time and is only related to the road conditions, the velocity-assisted initial alignment in motion can obtain higher alignment accuracy than the pure SINS navigation.

Since the OD outputs velocity information in the form of mileage increment, it can also directly adopt the form of displacement increment to assist in completing the initial alignment in motion, and the state equation of its Kalman filtering model is the same as that of Equation (26), differing only in the measurement equations. Please refer to reference [38] for details, which will not be repeated in this paper.

#### 2.3.3. Alignment in Motion Based on Carrier-Constrained

When the velocimetry device malfunctions, the velocity information provided by it is no longer available, and the alignment accuracy cannot be guaranteed without the assistance of velocity information and relying only on the SINS itself to complete the alignment in motion. Considering the actual driving situation of the vehicle on the road, the velocimetry device provides forward (*y*-axis) velocity information. If there is no lateral skidding and bumps and jumps and other phenomena, the vehicle’s normal (*z*-axis) and lateral (*x*-axis) velocity should be zero, which can be used as a measure to construct the Kalman filtering model when the velocimetry device is unable to provide the forward speed. This is called carrier constraints, also known as the dynamic zero velocity constraint or non-holonomic constraints (NHC).

Remove the equipment parameters of the velocimetry device in the state vector of the Kalman filtering model and construct the state vector as $X_{2}={\left[\begin{array}{ccccc} {\left(\phi^{n}\right)}^{T} & {\left(\delta v^{n}\right)}^{T} & {\left(\delta p^{n}\right)}^{T} & {\left(\varepsilon^{s}\right)}^{T} & {\left(\nabla^{s}\right)}^{T} \end{array}\right]}^{T}$, and state equation is as follows:(27)$$\left\{\begin{array}{ccc} {\dot{\phi }}^{n} & = & -\omega_{in}^{n}\times \phi^{n}+M_{dav}\delta v^{n}+M_{dap}\delta p^{n}-C_{s}^{n}\varepsilon^{s}-C_{s}^{n}w_{g}^{s}\\ \delta {\dot{v}}^{n} & = & f^{n}\times \phi^{n}+\left[v^{n}\times M_{dav}-2\omega_{ie}^{n}\times -\omega_{en}^{n}\times \right]\delta v^{n}+v^{n}\times M_{dvp}\delta p^{n}+C_{s}^{n}\nabla^{s}+C_{s}^{n}w_{a}^{s}\\ \delta {\dot{p}}^{n} & = & M_{dpv}\delta v^{n}+M_{dpp}\delta p^{n}\\ {\dot{\varepsilon }}^{s} & = & 0\\ \nabla^{s} & = & 0 \end{array}\right.$$

Rewriting the above equation in matrix form yields the following:(28)$${\dot{X}}_{2}=A_{2}X_{2}+B_{2}w_{2}$$
where $A_{2}=\left[\begin{array}{ccccc} -\omega_{in}^{n}\times & M_{dav} & M_{dap} & -C_{s}^{n} & 0_{3\times 3}\\ f^{n}\times & v^{n}\times M_{dav}-2\omega_{ie}^{n}\times -\omega_{en}^{n}\times & v^{n}\times M_{dvp} & 0_{3\times 3} & C_{s}^{n}\\ 0_{3\times 3} & M_{dpv} & M_{dpp} & 0_{3\times 3} & 0_{3\times 3}\\ 0_{6\times 3} & 0_{6\times 3} & 0_{6\times 3} & 0_{6\times 3} & 0_{6\times 3} \end{array}\right]$, $B_{2}=\left[\begin{array}{ccc} -C_{s}^{n} & 0_{3\times 3} & 0_{3\times 9}\\ 0_{3\times 3} & C_{s}^{n} & 0_{3\times 9}\\ 0_{9\times 3} & 0_{9\times 3} & 0_{9\times 9} \end{array}\right]$, $w_{2}={\left[\begin{array}{ccc} {\left(w_{g}^{s}\right)}^{T} & {\left(w_{a}^{s}\right)}^{T} & 0_{1\times 9} \end{array}\right]}^{T}$.

It is considered that the carrier constraints are described in the carrier coordinate system when constructing the measurement matrix, so transforming the velocities obtained from SINS calculations to the *b* system and taking the first-order minima yields the following:(29)$$\begin{array}{c} {\hat{v}}^{b}={\hat{C}}_{n}^{b}{\hat{v}}^{n}=C_{n}^{b}\left(I+\phi^{n}\times \right)\left(v^{n}+\delta v^{n}\right)\\ \approx v^{b}+C_{n}^{b}\delta v^{n}-C_{n}^{b}\left(v^{n}\times \right)\phi^{n} \end{array}$$(30)$$\delta v^{b}=-C_{n}^{b}\left(v^{n}\times \right)\phi^{n}+C_{n}^{b}\delta v^{n}$$

Measurement ***Z***_2_ is taken as the following:(31)$$Z_{2}=\left[\begin{array}{c} \delta v_{x}^{b}\\ \delta v_{z}^{b} \end{array}\right]=H_{2}X_{2}+u_{2}$$
where $u_{2}$ is measurement noise.

Setting $C_{n}^{b}=\left[\begin{array}{ccc} C_{11} & C_{12} & C_{13}\\ C_{21} & C_{22} & C_{23}\\ C_{31} & C_{32} & C_{33} \end{array}\right]$ and considering Equation (30), the measurement matrix can be obtained as follows:(32)$$H_{2}=\left[\begin{array}{ccccccc} C_{13}v_{N}^{n}-C_{12}v_{U}^{n} & C_{11}v_{U}^{n}-C_{13}v_{E}^{n} & C_{12}v_{E}^{n}-C_{11}v_{N}^{n} & C_{11} & C_{12} & C_{13} & 0_{1\times 9}\\ C_{33}v_{N}^{n}-C_{32}v_{U}^{n} & C_{31}v_{U}^{n}-C_{33}v_{E}^{n} & C_{32}v_{N}^{n}-C_{31}v_{N}^{n} & C_{31} & C_{32} & C_{33} & 0_{1\times 9} \end{array}\right]$$

The alignment in motion based on carrier constraints is modeled from Equations (28) and (31), and the initial alignment in the actual parking state can be considered as a special case of this method, which is just extended from two-direction carrier constraints in motion to three-direction carrier constraints in parking.

## 3. Vehicle-Mounted Alignment in Motion Based on Multi-Source Information Fusion

The dependence of GNSS-assisted on external information seriously affects the autonomy and stealth of special vehicles. Comprehensive and efficient utilization of the available information from multiple vehicle-mounted autonomous devices is an effective method to improve the performance of autonomous alignment for special vehicles in motion. In view of this, this paper proposes a vehicle-mounted alignment in motion method based on multi-source information fusion of federal Kalman filter (FKF) and provides a flexible and adaptive alignment strategy that combines the traveling or stopping state of the vehicle so that fault-tolerant filtering design can be realized when the auxiliary equipment fails to provide effective auxiliary information, and fault detection isolation and system reconfiguration can be accomplished to ensure the system’s normal operation under different operating conditions.

### 3.1. Alignment in Motion Based on Federal Kalman Filter

There are centralized Kalman filters and decentralized Kalman filters for information fusion using Kalman filters. The former centralizes all the information through a main filter to give the optimal estimation of the state, which has a high state dimension, a large computational cost, and poor fault tolerance. The latter is represented by the FKF proposed by Carlson, which has high accuracy, small computation, and good fault tolerance. In this paper, the FKF technique is used to fuse the SINS, OD, and LDV measurement information to realize the autonomous fine alignment in motion of optimal estimation.

#### 3.1.1. The Design of Federal Kalman Filter

The federal filter has been selected as the basic algorithm by the U.S. Air Force’s fault-tolerant navigation system “public Kalman filter” program [39], which can improve the accuracy and fault tolerance of the external auxiliary information. The federal filter generally adopts a two-stage filter structure. For alignment in motion, SINS is selected as the common reference system, and two first-stage sub-filter channels are formed with OD and LDV, respectively, and the local optimal estimation of the SINS states ${\hat{X}}_{i}$ and $P_{ii}$ (*i* = 1, 2) are given, and then the global estimation of the SINS states ${\hat{X}}_{g}$ and $P_{g}$ are obtained by the second-stage main filter fusion filtering. The information content of the state equation is inversely proportional to its process noise covariance, the information content of the state initial value is inversely proportional to its covariance array, and the SINS state equation and state initial value exist in each sub-filter and main filter, so there is a problem of information reuse. Considering the information conservation principle, the process noise information $Q^{-1}$ and the initial state estimation information $P^{-1}$ need to be allocated to all sub-filters according to the information allocation principle, which can be obtained as follows:(33)$$Q^{-1}=\sum_{i=1}^{n}Q_{i}^{-1}+Q_{m}^{-1}=\sum_{i=1}^{n}\beta_{i}Q^{-1}+\beta_{m}Q^{-1}$$(34)$$P^{-1}=\sum_{i=1}^{n}P_{i}^{-1}+P_{m}^{-1}=\sum_{i=1}^{n}\beta_{i}P^{-1}+\beta_{m}P^{-1}$$

Where *n* is the number of sub-filters, *i* represents the *i*th sub-filter, and *m* represents the main filter. The $\beta_{i}$ (*i* = 1,2, ……, *n* and *m*) is the information distribution coefficient and satisfies the following:(35)$$\sum_{i=1}^{n}\beta_{i}+\beta_{m}=1,(0\leq \beta_{i}\leq 1,0\leq \beta_{m}\leq 1)$$

In addition to satisfying the information conservation principle of Equation (35), the selection of $\beta_{i}$ should also consider the accuracy of each sub-filter. When the accuracy of a sub-filter is higher, a smaller $\beta_{i}$ should be set to ensure that the accuracy of that sub-filter is less affected by $\beta_{i}$, and the global estimation results are fed back to a larger share of the high-precision sub-filter. In the practical use of special vehicle alignment in motion, considering that OD and LDV have their own advantages and disadvantages, which are applicable to different road conditions, respectively, and it is not possible to simply specify who has a better auxiliary effect. Thus, this paper selects the same information allocation coefficients for the two sub-filter channels. In addition, the difference of $\beta_{i}$ also determines that the FKF has different structural forms, which can be designed according to the system requirements to obtain the corresponding fusion accuracy or fault tolerance, which is also a reflection of its design flexibility. Considering the reliability needs of alignment in motion of special vehicles, this paper adopts the non-reset mode federal Kalman filter with the best fault-tolerant performance of the main filter without information allocation [40], whose structure is shown in Figure 1, and the fault detection and isolation (FDI) link is further added subsequently to provide higher fault-tolerance performance.

Carlson proves that the global estimate of the federal filter is the optimal estimate provided that the state estimates of the sub-filters are uncorrelated ($P_{ij}=0,i\neq j$), at which point the globally optimal estimate is(36)$$\left\{\begin{array}{l} {\hat{X}}_{g}=P_{g}\sum_{i=1}^{n}P_{ii}^{-1}{\hat{X}}_{i}\\ P_{g}={\left(\sum_{i=1}^{n}P_{ii}^{-1}\right)}^{-1} \end{array}\right.$$
where ${\hat{X}}_{g}$ and $P_{g}$ are the global state estimates of the federal Kalman filter and its mean square array output. It can be seen that the worse the estimation effect of ${\hat{X}}_{i}$ (the larger $P_{ii}$), the smaller the share of ${\hat{X}}_{g}$ in the global estimation. However, each sub-filter of the initial alignment in motion is correlated ($P_{ij}\neq 0,i\neq j$), so it is necessary to amplify the process noise Q and the initial state covariance array $P_{ii}(t_{0})$ by a factor of $1/\beta_{i}$ times by means of the principle of information allocation and the technique of upper bound on the variance, where $\beta_{i}$ is the information allocation coefficients in Equation (35). At this time, the correlation term of each sub-filter can be ignored, which satisfies $P_{ij}=0,i\neq j$. After setting the process noise covariance and initial covariance of each sub-filter to be $1/\beta_{i}$ times of the whole filter, each sub-filter is measured and updated separately, and finally, the local estimation is fused according to Equation (36) to realize the global optimal estimation.

#### 3.1.2. The Design of Sub-Filter

The federal Kalman filter uses SINS as a common reference system with a state vector taken as $X_{SINS}={\left[\begin{array}{ccccc} {\left(\phi^{n}\right)}^{T} & {\left(\delta v^{n}\right)}^{T} & {\left(\delta p^{n}\right)}^{T} & {\left(\varepsilon^{s}\right)}^{T} & {\left(\nabla^{s}\right)}^{T} \end{array}\right]}^{T}$. According to the optimal estimation model developed in Section 2.3, there are(37)$${\dot{X}}_{\mathrm{SINS}}=A_{\mathrm{SINS}}X_{\mathrm{SINS}}+B_{\mathrm{SINS}}w_{\mathrm{SINS}}$$
where $A_{SINS}=\left[\begin{array}{ccccc} -\omega_{in}^{n}\times & M_{dav} & M_{dap} & -C_{s}^{n} & 0_{3\times 3}\\ f^{n}\times & v^{n}\times M_{dav}-2\omega_{ie}^{n}\times -\omega_{en}^{n}\times & v^{n}\times M_{dvp} & 0_{3\times 3} & C_{s}^{n}\\ 0_{3\times 3} & M_{dpv} & M_{dpp} & 0_{3\times 3} & 0_{3\times 3}\\ 0_{6\times 3} & 0_{6\times 3} & 0_{6\times 3} & 0_{6\times 3} & 0_{6\times 3} \end{array}\right]$, $B_{\mathrm{SINS}}=\left[\begin{array}{ccc} -C_{s}^{n} & 0_{3\times 3} & 0_{3\times 9}\\ 0_{3\times 3} & C_{s}^{n} & 0_{3\times 9}\\ 0_{9\times 3} & 0_{9\times 3} & 0_{9\times 9} \end{array}\right]$, $w_{2}={\left[\begin{array}{ccc} {\left(w_{g}^{s}\right)}^{T} & {\left(w_{a}^{s}\right)}^{T} & 0_{1\times 9} \end{array}\right]}^{T}$.

Sub-filter 1

Let $\delta K_{\alpha 1}={\left[\begin{array}{ccc} \delta K_{b1} & \delta \alpha_{x1} & \delta \alpha_{z1} \end{array}\right]}^{T}$ be the scale factor and mounting angle error of OD, $\delta L_{OD}$ be the rod arm vector error, and take $X_{OD}={\left[\begin{array}{cc} {\left(\delta K_{\alpha 1}\right)}^{T} & {\left(\delta L_{OD}\right)}^{T} \end{array}\right]}^{T}$. The state vector of sub-filter 1 is $X_{SINS/OD}={\left[\begin{array}{cc} {\left(X_{SINS}\right)}^{T} & {\left(X_{OD}\right)}^{T} \end{array}\right]}^{T}$. The model is referred to in Section 2.3.2, and the state equation can be rewritten as(38)$$\left[\begin{array}{c} {\dot{X}}_{SINS}\\ {\dot{X}}_{OD} \end{array}\right]=\left[\begin{array}{cc} A_{SINS} & 0_{15\times 6}\\ 0_{6\times 15} & A_{OD} \end{array}\right]\left[\begin{array}{c} X_{SINS}\\ X_{OD} \end{array}\right]+\left[\begin{array}{cc} B_{SINS} & 0_{15\times 6}\\ 0_{6\times 15} & B_{OD} \end{array}\right]\left[\begin{array}{c} w_{\mathrm{SINS}}\\ w_{\mathrm{OD}} \end{array}\right]$$
where $A_{\mathrm{OD}}=0_{6\times 6}$, $B_{\mathrm{OD}}=0_{6\times 6}$, $w_{\mathrm{OD}}=0_{1\times 6}$.

The difference between the SINS calculated velocity and the OD velocity was used as the measurement $Z_{\mathrm{SINS}/\mathrm{OD}}$, and the measurement equation was constructed as(39)$$Z_{\mathrm{SINS}/\mathrm{OD}}=H_{\mathrm{SINS}/\mathrm{OD}}X_{\mathrm{SINS}/\mathrm{OD}}+u_{\mathrm{SINS}/\mathrm{OD}}$$
where the measurement matrix is as follows:(40)$$H_{\mathrm{SINS}/\mathrm{OD}}=\left[\begin{array}{cccc} -v^{n}\times & I_{3\times 3} & 0_{3\times 3} & \begin{array}{cccc} -C_{s}^{n}\left(L_{\mathrm{OD}}\times \right) & 0_{3\times 3} & -C_{b}^{n}M_{v} & C_{s}^{n}\left(\omega_{eb}^{s}\times \right) \end{array} \end{array}\right]$$

2.Sub-filter 2

Let $\delta K_{\alpha 2}={\left[\begin{array}{ccc} \delta K_{b2} & \delta \alpha_{x2} & \delta \alpha_{z2} \end{array}\right]}^{T}$ be the scale factor and mounting angle error of LDV, $\delta L_{\mathrm{LDV}}$ be the rod arm vector error, and take $X_{LDV}={\left[\begin{array}{cc} {\left(\delta K_{\alpha 2}\right)}^{T} & {\left(\delta L_{LDV}\right)}^{T} \end{array}\right]}^{T}$. The state vector of sub-filter 2 is $X_{\mathrm{SINS}/\mathrm{LDV}}={\left[\begin{array}{cc} {\left(X_{\mathrm{SINS}}\right)}^{T} & {\left(X_{\mathrm{LDV}}\right)}^{T} \end{array}\right]}^{T}$. The state and measurement equations are similar to Equations (38) and (39), except that the OD subscript is changed to LDV.

### 3.2. Fault-Tolerant Design of Alignment in Motion Based on Multi-Source Information Fusion

The fault-tolerant design of alignment in motion based on the multi-source information fusion can be divided into coarse and fine alignment phases, which are fault-tolerant designs for inertial system alignment and federated filtering optimal estimation, respectively.

#### 3.2.1. Fault-Tolerant Designs for Inertial System Alignment

Coarse alignment in motion of an inertial system uses OD and LDV to measure the carrier velocity, respectively, and compensates for the motion interference of acceleration to obtain the gravity vector under the inertial system and complete the alignment of the inertial system. In the process of vehicle driving, due to the complex road conditions, wheel slip, idling, and other conditions will cause the OD velocity measurement error. The road surface’s concave and convex amplitude or the existence of water surface ice will cause the depth of field changes in the LDV velocimetry process, resulting in reflective signal noise being larger and the same bringing velocity measurement error. In addition, ODs and LDVs may also experience equipment failures such as communication blackouts, resulting in the equipment not functioning properly. These velocimetry errors or failures will affect the gravity vector solving accuracy during the inertial system alignment process, leading to an increase in the alignment error, which requires the addition of a fault-tolerant design to enhance the reliability of inertial system alignment in motion.

The accelerometer in the IMU is solidly connected to the vehicle body, and the output is the velocity increment during the sampling time interval, which can be calibrated to obtain the forward velocity increment of the vehicle, and the forward velocity $v_{b}$ of the vehicle can be obtained through integration. Therefore, the information about the forward velocity of the vehicle is redundant in the OD, LDV, and accelerometer outputs, where the IMU has the highest reliability and both the OD and the LDV can fail. Therefore, the forward velocity of the vehicle measured by the IMU can be used to detect whether the speeds measured by the OD and LDV are incorrect or not, so as to realize the fault diagnosis and isolation of the velocity measurement auxiliary equipment. Generally speaking, it is rare for OD and LDV to fail at the same time. When one of OD or LDV fails or has a large error in the measured value, the forward speed of the vehicle solved by IMU can be compared with the measured velocity $v_{od}$ of OD and $v_{ldv}$ of LDV, respectively, and the one with a smaller absolute value of the difference will be the usable velocity measurement channel, and then isolate the faulty channel after diagnosis and continue to assist in the inertial system alignment with the use of the other velocity measurement channel. The fault-tolerant design is shown in Figure 2.

It should be noted that the probability of simultaneous failure of OD and LDV is very small, and even if both fail at the same time, if the failure time accounts for a small portion of the total alignment time, converged alignment curves can still be obtained, with only a slight increase in the alignment error. This is due to the fact that the coarse alignment in motion is in the form of inertial system vector integration, which can smooth a small amount of disturbance data over the overall alignment result and has a certain capacity to resist disturbance, a conclusion that will be verified by experiments in Section 4.3.

#### 3.2.2. Fault-Tolerant Design of Optimal Estimation Alignment Based on FKF

The inertial navigation system is highly reliable and generally does not fail, otherwise, the initial alignment of the SINS cannot be completed, so the FDI is mainly for the OD and LDV channels. The fault-tolerant design of the system includes fault detection and reconfiguration of the faulty system, so that when one of the devices fails, fault detection and isolation can be realized and the other normal working channels can be used to reconfigure the filters to ensure that the alignment process proceeds normally. When the sensor faults are removed, the sub-system can be restored to the federal filter.

Fault detection and isolation

State test and residual test are two commonly used fault detection methods based on the $\chi^{2}$ test, in which the state test method utilizes the difference between the state estimation ${\hat{X}}_{k}$ that contains measurement information and the state estimation ${\hat{X}}_{k}^{S}$ that does not contain measurement information to complete the test and isolation of the measurement information validity. The disadvantage of ${\hat{X}}_{k}^{S}$ is that it is affected by the initial value and the model noise, the sensitivity is lower and the computational amount is larger. Therefore, this paper selects the residual test as the fault diagnosis and isolation method.

Let the measurement residual be(41)$$r_{k}=Z_{k}-H_{k}{\hat{X}}_{k,k-1}$$

When operating normally, the residual $r_{k}$ obtained from the measurement $Z_{k}$ and the measurement model extrapolation $H_{k}{\hat{X}}_{k,k-1}$ satisfies $r_{k}\sim N\left(0,C_{r_{k}}\right)$, where $C_{r_{k}}=H_{k}P_{k,k-1}H_{k}^{T}+R_{k}$. When a fault occurs, the residual $r_{k}$ has a non-zero mean value. Therefore, the following binary assumption can be made:

Without fault H_0_: $E\left[r_{k}\right]=0$, $E\left[r_{k}r_{k}^{T}\right]=C_{r_{k}}$;

With fault H_1_: $E\left[r_{k}\right]=\mu \neq 0$, $E\left\{\left[r_{k}-\mu \right]{\left[r_{k}-\mu \right]}^{T}\right\}=C_{r_{k}}$;

Define the fault test function as(42)$$\lambda_{k}=r_{k}^{T}C_{r_{k}}^{-1}r_{k}$$

When the work is normal, it satisfies $\lambda_{k}\sim \chi^{2}\left(m\right)$, *m* = 3 as the dimension of measurement. Once a fault occurs then $\lambda_{k}\sim \chi^{2}\left(3\right)$ is not satisfied, so the test threshold $T_{\lambda }$ is selected and the fault judgment criterion is(43)$$\left\{\begin{array}{l} \lambda_{k}\leq T_{\lambda },faulted\\ \lambda_{k}>T_{\lambda },fault\;\text{-}\;free \end{array}\right.$$
where $T_{\lambda }$ is the upper ζ quantile of the $\chi^{2}\left(3\right)$ distribution and ζ is the false alarm rate.

2.Reconfiguration of system

When the system equipment is working normally, fusion estimation of the federal Kalman filter is performed by sub-filters 1 and 2 after filtering separately. When an auxiliary velocimetry device fails, the measurement information of this channel is masked, and another sub-filter continues to perform Kalman filtering separately. When both OD and LDV failures are detected, all the corresponding sub-filters should be isolated and transferred to the carrier-constrained auxiliary INS described in Section 2.3.3 to complete the initial alignment, and then the filtering channel is resumed after the auxiliary device returns to normal. Corresponding faults and system reconfiguration schemes are shown in Table 1.

### 3.3. Alignment in Motion of Vehicle-Mounted SINS

#### 3.3.1. Information Multiplexing for Alignment in Motion

Whether initial alignment is in wobble base or in motion, the same convergence performance can obtain higher alignment accuracy in a longer alignment time. If the fine alignment stage can reuse the data information of the coarse alignment stage, it is equivalent to extending the time, and thus better convergence effect and alignment accuracy can be obtained. Based on the above idea, for the initial alignment of the shaking base, ignoring the small change of attitude caused by the shaking interference, the attitude result at the end of the coarse alignment stage is basically the same as the alignment start moment, so the fine alignment stage can be advanced to the whole alignment start moment, and the result of the coarse alignment can be used as the initial attitude of the fine alignment. However, unlike the wobble base alignment in the parking state, the carrier attitude changes at any time during the alignment process in motion, so the above information multiplexing method cannot be used. Considering that the idea of inertial system alignment is to solidify the carrier system and the inertial system into the inertial space at the initial moment and solve the attitude conversion matrix between them by using the vectorial attitude fixing method, the attitude at the beginning of alignment can be deduced inversely from the results of inertial system alignment and then be used as the initial value of the attitude to start the optimal estimation of the fine alignment.

The detailed process of deducing inversely is as follows. Since the inertial system method is used in the coarse alignment stage, $C_{i_{0}}^{n_{0}}$ of Equation (1) at the starting moment can be obtained from Equation (3) (the position has not yet changed), and $C_{i_{b0}}^{i_{0}}$ can be obtained from Equation (9). As for $C_{b}^{i_{b0}}$ of Equation (1), the definition of *i_b_*_0_ shows that there is $C_{b}^{i_{b0}}=I_{3\times 3}$ at the starting moment of alignment, so the rough attitude at the starting moment of alignment can be obtained from $C_{i_{b0}}^{i_{0}}$ in the result of inertial system coarse alignment in motion and the initial geographic position information, which can be used as the initial value to carry out the optimal estimation of fine alignment in motion. At the same time, considering the large interference of alignment in motion, the coarse alignment time can be extended in order to obtain usable coarse alignment results, and the diagram of alignment in motion based on information multiplexing is shown in Figure 3.

Adopting the above alignment in motion method based on information multiplexing not only increases the alignment time and improves the alignment accuracy, but also can update the attitude, velocity, and position information of the carrier in real time from the starting moment of alignment according to the strapdown inertial navigation algorithm in the process of fine alignment. In addition, although the information multiplexing method cannot give the current moment attitude in real time when the inertial system is transferred from coarse alignment to fine alignment and it is necessary to carry out fine alignment on the historical data to obtain the attitude information, considering the processing speed of the current high-performance computers, the fine alignment solving of the historical data for a shorter period of time (about 10 min) can be completed very quickly. In the process of solving, the output data of SINS and auxiliary equipment can be recorded continuously, which can “catch up” to the real-time attitude update and not lose data.

#### 3.3.2. Adaptive Alignment Strategy of Vehicle-Mounted SINS

Special vehicles do not travel immediately after powering up in practice and often require a few minutes of state preparation while parking. Therefore, there are two simple initial alignment strategies. Strategy 1 is to use the wobble base coarse alignment in the inertia system plus optimal estimation fine alignment in the parking state, which is also the most widely used option at present. Strategy 2 is to use coarse alignment in the inertial system plus optimal estimation fine alignment in motion, regardless of whether the vehicle is traveling or not.

Comparing the two initial alignment schemes, Strategy 1 utilizes the actual ground speed of the carrier to be zero and avoids the influence of the output error of the velocimetry equipment and the model error on the initial alignment accuracy under the parking state and also improves the initial alignment accuracy through low-pass filtering and rotational modulation, but it cannot be carried out normally after the vehicle travels. Strategy 2 utilizes the auxiliary equipment to measure the velocity and compensate for the motion interference, which can realize the initial alignment under the vehicle traveling state, but the error of the velocimetry equipment has a large influence on the results. Taking full advantage of the benefits of both schemes, this paper proposes an adaptive alignment strategy for vehicle-mounted SINS, which utilizes the output data of the velocimetry device to determine whether the carrier is parked or not, adopting Strategy 1 if parked and switching from Strategy 1 to Strategy 2 if the carrier enters the traveling state. The adaptive alignment strategy is shown in Figure 4. The following description is provided exclusively for the adaptive alignment technique.

The outputs of OD and LDV are used to determine whether the vehicle is moving or not. In the parking state, the output velocities of OD and LDV are anticipated to be zero. With the idea of fault detection, let the output velocity $v_{k}$ of OD and LDV in the parking state be zero-mean Gaussian white noise ($v_{k}∼N\left(0,C_{v_{k}}\right)$), where $C_{v_{k}}$ is the variance of the output of the velocimetry device. When the vehicle starts and travels, the mean value of $v_{k}$ will change. Therefore, the binary assumption is made as follows:

Parking state H_0_: $E\left[v_{k}\right]=0$, $E\left[v_{k}v_{k}^{T}\right]=C_{v_{k}}$.

Vehicle travel H_1_: $E\left[v_{k}\right]=\mu_{v}\neq 0$, $E\left\{\left[v_{k}-\mu_{v}\right]{\left[v_{k}-\mu_{v}\right]}^{T}\right\}=C_{v_{k}}$.

Define the fault test function as(44)$$\lambda_{v_{k}}=v_{k}^{T}C_{v_{k}}^{-1}v_{k}$$

In the parking state, the fault test function $\lambda_{v_{k}}$ obeys the χ^2^ (2) distribution (Consider both OD and LDV forward velocities, so the dimension is 2). When the vehicle is moving, it no longer obeys the χ^2^ (2) distribution. Therefore, the test threshold is taken as ${T_{\lambda }}^{\prime }$, and the fault judgment criterion is selected as(45)$$\left\{\begin{array}{l} \lambda_{v_{k}}\leq {T_{\lambda }}^{\prime },Stop\\ \lambda_{v_{k}}>{T_{\lambda }}^{\prime },Move \end{array}\right.$$
where ${T_{\lambda }}^{\prime }$ is the upper side $\zeta^{\prime }$ quantile of the χ^2^ (2) distribution and $\zeta^{\prime }$ is the false alarm rate.

The initial alignment process is switched using a time threshold. After initialization to determine whether the vehicle has been in the parking state; if so, then Strategy 1 is used: the wobble base coarse alignment in the inertia system of 1 min, and increase the multi-vector integration and low-pass filtering means to inhibit the impact of high-frequency vibration. If the parking state is more than 1 min, then into the wobble base fine alignment of optimal estimation based on the Kalman filter, and increase the rotational modulation to inhibit the impact of the inertial device constant value error and improve the accuracy of the alignment.

Once the special vehicle is out of the parking state, it enters the initial alignment in motion. After entering the alignment in motion process, determine whether the coarse alignment of the wobble base before transferring to the traveling state has been completed (judging by the time, the coarse alignment of the wobble base is considered to be completed if it is more than 1 min, corresponding to *flag* = 1 in Figure 4). If it is completed, then take the final result of the initial alignment of the wobble base as the initial value, and then transfer it to the fine alignment in motion with optimal estimation based on the Kalman filter. The information multiplexing method in Section 3.3.1 can be used until the cutoff time is reached to give the alignment result. If the coarse alignment of the wobble base is not completed before transferring to the traveling state (less than 1 min), the final result of the initial alignment of the wobble base is used as the initial value, and then transferring to the inertial system coarse alignment in motion and then carrying out the Kalman optimal estimation fine alignment in motion, the process can also be used in the same way as the information multiplexing method of Section 3.3.1 until the cutoff time is reached to give the alignment result.

## 4. Experiments and Discussion

In order to verify the validity of the vehicle-mounted alignment in motion method and fault-tolerant design based on multi-source information fusion proposed in this paper, an experiment is conducted by the vehicle-mounted SINS test system, which accumulates the measured data for validation. The vehicle-mounted SINS test system is shown in Figure 5, which includes SINS, GPS, OD, LDV, a barometric altimeter, test software, and other equipment. The INS is a dual-axis laser strapdown inertial navigation system, in which the IMU output frequency is 100 Hz, the gyroscope constant value drift is 0.005°/h and the random error is 0.0005°/h^1/2^, the accelerometer constant value zero bias is 50 μg and the random error is 5 μg/Hz^1/2^, the accuracy of the GPS receiver is 5 m and the working frequency is 1 Hz, the error of the OD scale factor is 0.2%, the accuracy of the LDV is ±0.05%, and barometric altimeter accuracy is 10 m.

The test vehicle first completes the initial alignment of the parking state at the starting point and then enters the state of SINS/GPS integrated navigation, and the navigation result is used as a reference benchmark to check the accuracy of other measurement schemes. The route of the test vehicle is shown in Figure 6, and the raw output data saved during the test are simulated offline, and the attitude and velocity of the SINS/GPS integrated navigation are shown in Figure 7.

### 4.1. Experiment and Validation of Alignment in Motion Based on Information Multiplexing

The above-measured data of the test vehicle are used to verify the effect of the information multiplexing method in the alignment in motion, taking the attitude output of the integrated SINS/GPS navigation after the initial alignment of the parking as the benchmark, taking the 10 min traveling state data, carrying out the inertial system alignment in motion firstly, and then deducing inversely the rough attitude at the beginning moment of the alignment, and then carrying out the LDV velocity-assisted, OD and LDV displacement incremental-assisted optimal estimation of fine alignment for the test data in different time periods of this test, respectively, and compared with the whole process in inertial frame assisted by LDV displacement increment. The test results are displayed in Figure 8. The measured data from different time periods of this test were aligned 10 times, and the results are shown in Table 2.

The results show that the alignment accuracy of the inertial system without information multiplexing for the whole 10 min is the worst, and the mean value of the azimuth error angle is about 53.31′, while the optimal estimation of the information multiplexing has a very significant improvement, in which the optimal estimation of the LDV velocity -assisted azimuth error angle is about 37.70′, the optimal estimation of the OD displacement increment-assisted azimuth error angle under the *b* system is about 6.97′, and the optimal estimation of the LDV displacement incremental-assisted under the *b* system has the highest alignment accuracy with a mean value of azimuth error angle of about −1.99′. This is because the scale factor of the LDV is more stable, and the alignment accuracy is better than that of the OD-assisted alignment in motion when the road surface is relatively flat and there is no reflection from the water surface and ice.

### 4.2. Experimental Validation of Alignment in Motion Based on Federal Kalman Filter

Alignment in motion based on the federal Kalman filter is performed on the measured data with the information allocation coefficients set to $\beta_{1}=\beta_{2}=1/2$ and $\beta_{m}=0$, and the non-reset mode federal Kalman filter structure described in Section 3.1.1 is adopted. The same inertial system coarse alignment in motion is performed first with the 10 min data, and the OD/LDV displacement incremental-assisted federal Kalman filter fine alignment under the *b* system is performed again with the information multiplexing method in Section 3.3.1. The main filter update time is 10 s, and the results are shown in Figure 9.

In order to further validate the results of alignment in motion based on the federal Kalman filter, OD/LDV displacement increment-assisted federal Kalman filter alignment in motion under the *b* system was performed 10 times at different time intervals of the measured data, and the optimal estimation of GPS-assisted SINS alignment after static base alignment was also used as a benchmark for comparison, and the results are shown in Table 3.

From Figure 9 and Table 3, it can be seen that alignment in motion based on the federal Kalman filter utilizes the information from both the OD and LDV channels in a combined manner and has a higher alignment accuracy compared to the separate OD and LDV carrier displacement increment-assisted optimal estimation alignment in Section 4.1, which is a strong proof of the validity and sophistication of the method proposed in this paper.

### 4.3. Experimental Validation of Fault-Tolerant Design Based on Multi-Source Information Fusion

#### 4.3.1. Validation of Fault-Tolerant Design in Motion Under Inertia System

In order to verify the effect of the fault-tolerant design in motion under the inertia system, the OD and LDV output data in the measured data are subject to simulated fault processing: the 200–210 s section of the OD output data is increased by five values to simulate the idling condition of the wheels of the special vehicles, i.e., the velocity of the vehicle is smaller than the OD output velocity, and the 350–360 s section is reduced by five values to simulate the wheel-slip condition of the special vehicles, i.e., the actual velocity of the vehicle is greater than the OD output velocity. The velocity data output by the LDV is reduced by 5000 values in the 350–360 s section and increased by 5000 values in the 500–510 s section, and the simulated faulty measured data is tested through the fault-tolerant design in Section 3.2.1.

The test results are displayed in Figure 10. The OD auxiliary inertia system alignment curve starts to show jitter at 200 s and 350 s, and the alignment error increases. The LDV auxiliary alignment curve starts to show jitter at 350 s and 500 s, and the inertia system alignment curve with the addition of the fault-tolerant design is not affected when the single auxiliary equipment fails at the two places of 200 s and 500 s. Due to the simultaneous failure of the two velocimetry devices at the place of 350 s, the curves were inevitably affected. The results demonstrated that the fault-tolerant design in motion under the inertial system given in Section 3.2.1 can diagnose the faulty data and channels, isolate the faulty velocimetry channels, switch the fault-free channels to avoid the alignment errors caused by individual velocimetry equipment failures, and provide sufficiently accurate coarse alignment results for the subsequent optimal estimation of the alignment in motion. It should be noted that, although both OD and LDV are faulted at 350 s, the fault time is very short compared with the whole alignment time of 10 min, and the multi-vectors integration method under the inertial system has a smoothing effect on a small amount of error, and the curves converge after the fault, which reflects the anti-interference ability of multi-vector integration alignment under the inertial system.

#### 4.3.2. Validation of Optimal Estimation Fault-Tolerant Design Based on FKF

In order to verify the capability of FDI based on the federal Kalman filter, simulated faults are carried out on the measured data, and the OD output data are verified to be consistent with the fault-tolerance design in motion under the inertial system. Since the optimal estimation uses the velocity information solved from the LDV accumulated mileage data as an auxiliary, the accumulated mileage data of the LDV is increased by 100 in the section of 350–360 s and decreased in the section of 500–510 s by 100, and the false alarm rate is set to 0.05. The simulated faulty measured data are tested by the fault-tolerant design in Section 3.2.2, and the results are shown in Figure 11, which show that several faults added by the simulation are effectively recognized, in which the OD and the LDV are faulted at 350–360 s. The data after the simulated faults were first subjected to multi-vector integration coarse alignment under the inertial system, and the results were used as the initial approximate attitude for optimal estimation of fine alignment in motion using the information multiplexing technique. For comparative validation, the optimal estimation alignment in motion with separate assistance of OD and LDV was performed, and the results are shown in Figure 12.

It can be seen from Figure 12 that the OD-assisted optimal estimation alignment curves are jittery at both 200 s and 350 s, and the LDV-assisted optimal estimation alignment curves are jittery at both 350 s and 500 s and although the curves gradually converge with subsequent correctly measured data-assisted filtering estimation, the alignment results have a large error. The federal Kalman filter alignment is performed by the fault-tolerant design in Section 3.2.2, and the experimental results are shown in Figure 13.

As seen in Figure 13, the FDI based on the federal Kalman filter effectively diagnoses and isolates the faults at 200 s, 350 s, and 500 s, and has more accurate alignment results, the detailed operation mechanism of which is as follows. When the FDI determines that the OD and LDV measurement data are normal ($\lambda_{k}\leq T_{\lambda }$), the alignment in motion adopts optimal estimation of the federal filter through the fusion of the two sub-filters of OD and LDV in Section 3.1.1. When a large measurement residual is detected in the OD channel ($\lambda_{k}>T_{\lambda }$), the channel is isolated, and the alignment in motion of optimal estimation is performed solely through the LDV. Similarly, when a large measurement residual is detected in the LDV channel, the channel is isolated, and the alignment in motion of optimal estimation is performed solely through the OD. When faults are diagnosed in both measurement channels, the carrier-constrained optimal estimation in motion is enabled, reducing the impact of the auxiliary equipment faults on the filtering process and improving the alignment accuracy and fault-tolerance capability of the initial alignment system.

## 5. Conclusions

The initial alignment in motion can provide support for the environmental adaptation and rapid mobility of special vehicles in special backgrounds, and the reliability, stability, and autonomy of the system in complex practical application environments are also key challenges that need to be solved. In this paper, a method of alignment in motion based on the multi-source information fusion from vehicle-mounted autonomous devices is proposed. On the one hand, the federal Kalman filtering fully integrates multiple external auxiliary information, which effectively provides more accurate initial alignment results. On the other hand, the two designed fault-tolerant schemes of the inertial system and federal filter optimal estimation comprehensively improve the performance of fault diagnosis and isolation of the initial alignment system. In addition, this paper gives a strategy of adaptive selection of alignment method according to the complex and multifarious actual working conditions, which provides a feasible idea to further improve the accuracy of alignment in motion and the actual performance of special vehicles. Finally, the effectiveness and feasibility of the proposed method are verified by experiments of the test vehicle, and the results show that the accuracy of the alignment in motion results is significantly improved by using the federal Kalman filter, and the randomly occurring simulated faults are accurately identified and successfully isolated, which effectively enhances the robustness and practical use performance of the initial alignment system.

Further, the availability of GNSS is diagnosed, whether or not through the redundant velocity information of OD, LDV, and GNSS, and adding the federal Kalman filter to further improve the alignment accuracy and fault tolerance when it is available will be further carried out in the future research work. In addition, increase the number of experiments under different road conditions and multiple maneuvering conditions, so as to fully verify the accuracy and robustness of the algorithm.

## Figures and Tables

**Figure 1 entropy-27-00237-f001:**
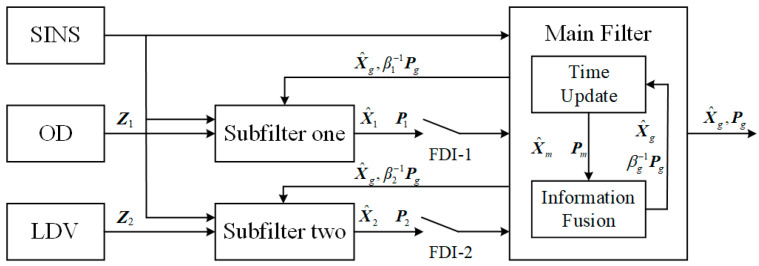
Structure of the federal Kalman filter design for multi-source information fusion.

**Figure 2 entropy-27-00237-f002:**
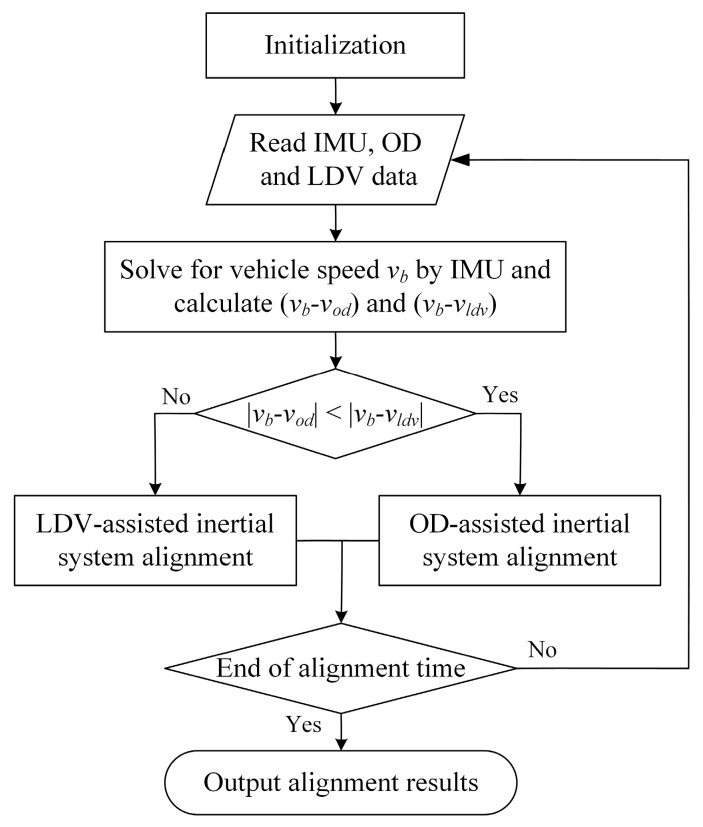
Fault-tolerant design of inertial system alignment in motion.

**Figure 3 entropy-27-00237-f003:**
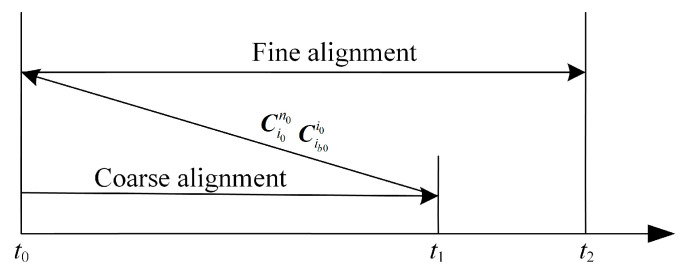
Schematic diagram of alignment in motion based on information multiplexing.

**Figure 4 entropy-27-00237-f004:**
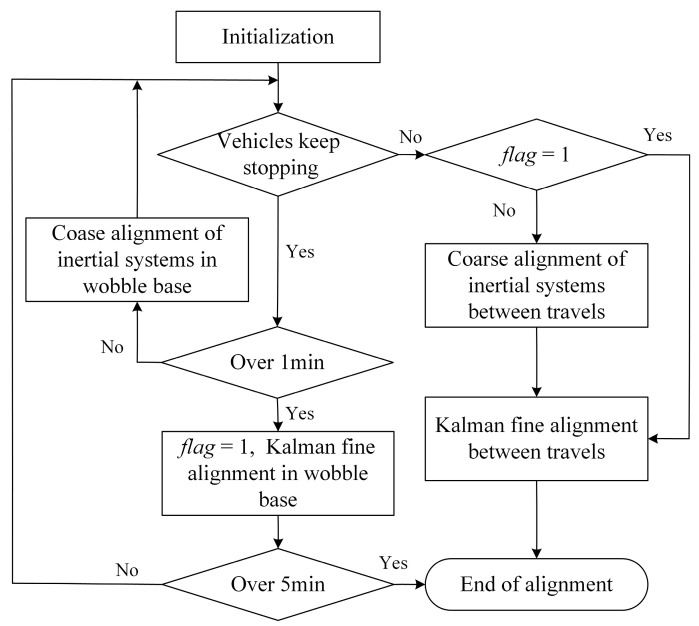
Flowchart of adaptive alignment strategy.

**Figure 5 entropy-27-00237-f005:**
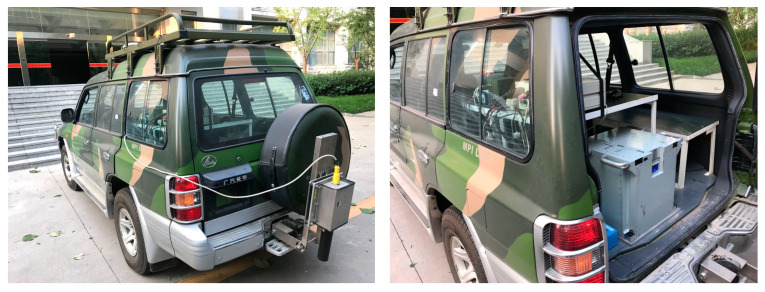
Experimental platform of vehicle-mounted SINS.

**Figure 6 entropy-27-00237-f006:**
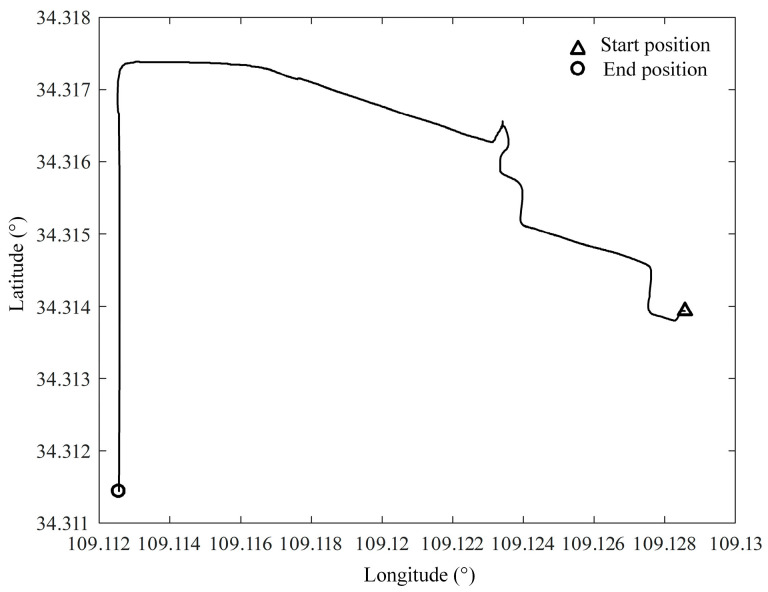
Experimental roadmap of test vehicle.

**Figure 7 entropy-27-00237-f007:**
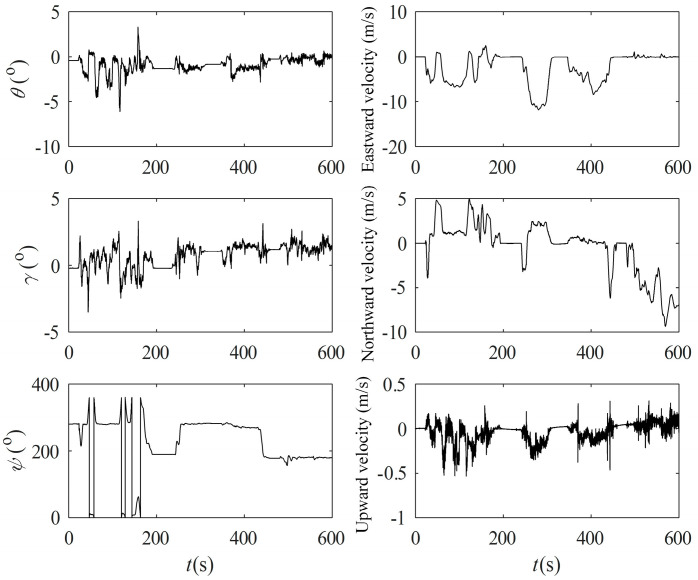
Variation of attitude and velocity.

**Figure 8 entropy-27-00237-f008:**
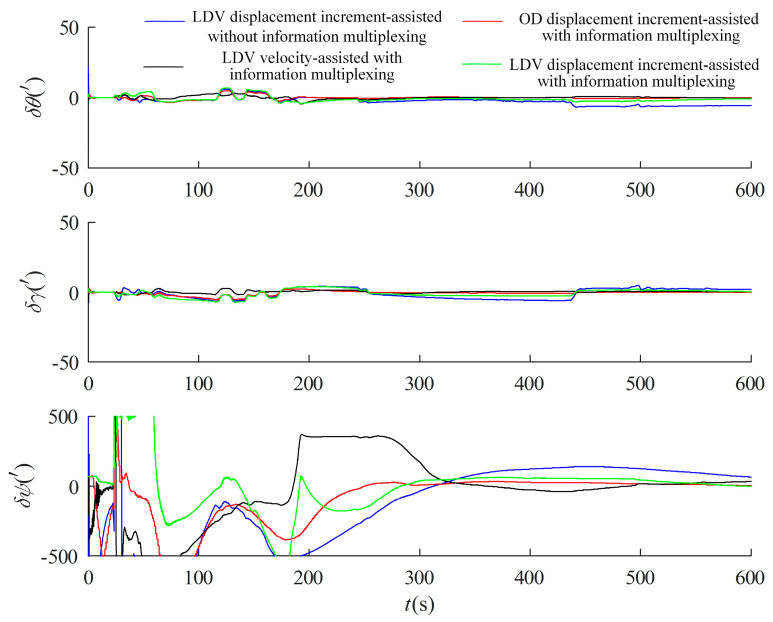
Experimental result of alignment in motion based on information multiplexing.

**Figure 9 entropy-27-00237-f009:**
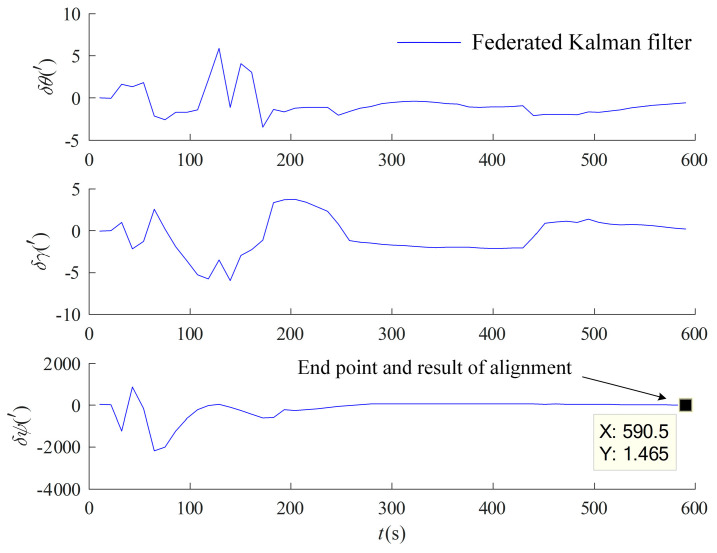
Alignment in motion based on OD/LDV federal Kalman filter.

**Figure 10 entropy-27-00237-f010:**
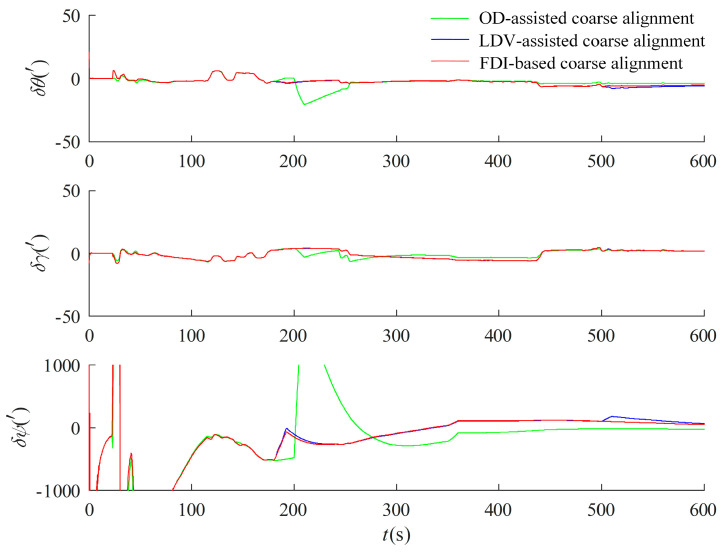
Test results of alignment fault-tolerant design with fault under inertial system.

**Figure 11 entropy-27-00237-f011:**
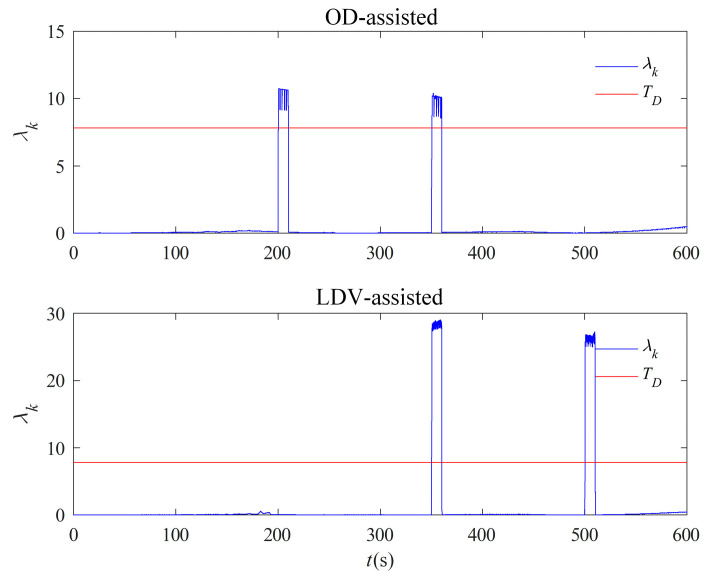
Fault detection results of OD and LDV auxiliary based on federal filter.

**Figure 12 entropy-27-00237-f012:**
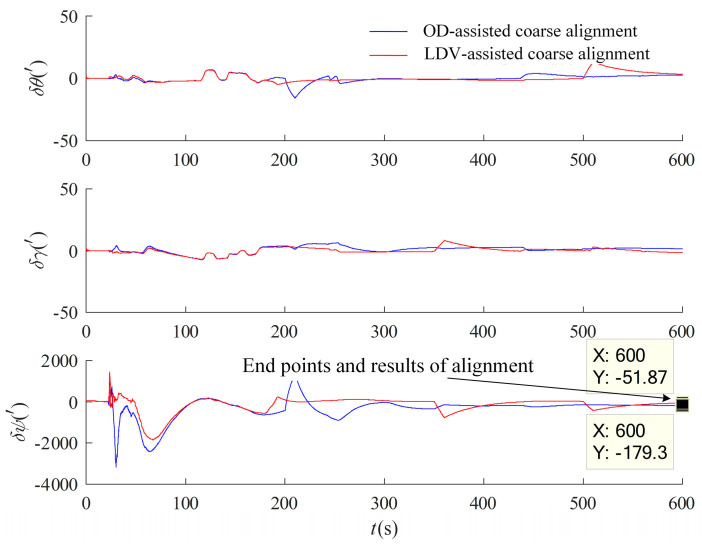
Result of optimal estimation alignment with faults of OD and LDV auxiliary.

**Figure 13 entropy-27-00237-f013:**
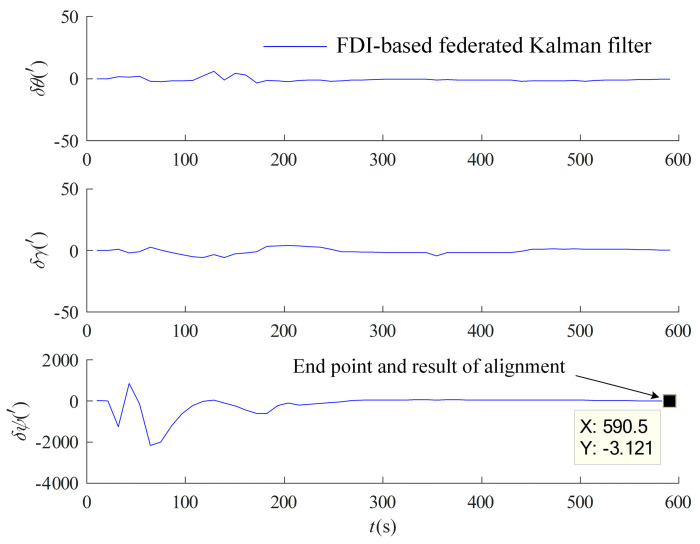
Result of optimal estimation alignment with faults of FDI based on federal filtering.

**Table 1 entropy-27-00237-t001:** Faults and system reconfiguration scheme.

System State	Reconfiguration Scheme	Output Filter
No faults	Sub-filters 1,2	Main filter
OD faults	Sub-filter 2	Main filter
LDV faults	Sub-filter 1	Main filter
OD and LDV faults	carrier-constrained	INS based on carrier-constrained
SINS faults	None	Stop alignment

**Table 2 entropy-27-00237-t002:** Azimuth error angle of 10 times alignment in motion with information multiplexing (′).

δ*ψ*	Without Information Multiplexing	With Information Multiplexing		
	LDV Displacement Incremental-Assisted	LDV Velocity-Assisted	OD Displacement Incremental-Assisted	LDV Displacement Incremental-Assisted
1	53.0033	34.1917	6.7325	−1.3236
2	53.2928	34.2004	6.4604	−1.0686
3	53.4124	35.6140	7.0388	−1.2772
4	53.1496	35.4842	6.8942	−1.3188
5	52.9976	37.9575	6.2910	−1.3428
6	53.2625	38.0623	7.1083	−1.1034
7	53.5906	40.4466	7.2228	−1.8044
8	53.5683	40.4644	6.7506	−2.2764
9	53.4982	40.4469	8.0432	−2.7803
10	53.3461	40.1758	7.1689	−5.6472
Mean	53.3121	37.7044	6.9711	−1.9943
1σ	0.2031	2.6431	0.4842	1.3973

**Table 3 entropy-27-00237-t003:** Test results of alignment in motion based on federal Kalman filter (′).

	Δθ	δγ	δψ
1	−0.5611	0.2080	1.4652
2	−0.5578	0.2083	1.3618
3	−0.5581	0.2075	1.3830
4	−0.5585	0.2064	1.4036
5	−0.5551	0.2067	1.2919
6	−0.5556	0.2057	1.3170
7	−0.5562	0.2043	1.3488
8	−0.5530	0.2046	1.2452
9	−0.5532	0.2039	1.2596
10	−0.5533	0.2029	1.2758
Mean	−0.5562	0.2058	1.3352
1σ	0.0027	0.0019	0.0702

## Data Availability

The data presented in this study are available on request from the corresponding author.

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
