# Peer review of "In-Motion Initial Alignment Method Based on Multi-Source Information Fusion for Special Vehicles"

_entropy, 2025, doi:10.3390/e27030237_

Round 1
Reviewer 1 Report
Comments and Suggestions for Authors
This document presents itself as a grand technical research report rather than an academic paper, leading to an overly extensive and diverse set of content that makes it challenging to identify the main points.
1. The abstract places excessive emphasis on vague concepts and grand statements, lacking a clear presentation of the methods or innovations discussed in the report.
2. The discussion on the alignment technology of inertial navigation systems is too broad. It would be more effective if the paper focused on a specific aspect, such as process adaptability, multi-source information fusion, or rotational alignment during movement.
3. In the adaptive alignment flowchart, when parking time is greater than 1 minute but less than 5 minutes, flag=0. Is it still necessary to conduct coarse alignment during movement in this case?
4. At the beginning of page 20, it is noted that the odometer output is zero under parking conditions, which is consistent with the static zero-velocity assisted alignment model. Why is there a need for additional verification for parking? Does the odometer-assisted optimal estimation alignment dynamic model not accommodate static conditions?
5. The validity of using the alignment results from the combination of GNSS and inertial navigation systems during movement as a reference benchmark for autonomous alignment should be reconsidered. The GNSS-aided alignment accuracy is significantly influenced by carrier maneuverability and GNSS speed accuracy. Given that rotational inertial navigation is utilized, would it not be more appropriate to use the static rotational alignment results at the end of the in-motion alignment as a reference?
6.There are missing units for the numerical values presented in Table 3, which should be addressed for clarity.
Author Response
Under the guidance of the reviewers, we have indeed found that the content of this manuscript is overly extensive and diverse, and does not focus enough on the technical main point of the alignment method based on multi-source information fusion and fault diagnosis. There is a certain gap between it and a qualified academic paper. We thank the reviewer for reviewing this manuscript, and the review comments were of significant help in improving the quality of the manuscript. The review comments were revised individually, and the point-to-point responses are shown below.
(1) The abstract places excessive emphasis on vague concepts and grand statements, lacking a clear presentation of the methods or innovations discussed in the report.
The author agrees with the professional and specific suggestions raised by the reviewing experts on the writing of the abstract. Based on this, we have removed vague concepts and grand statements, and briefly explained the background of the problem research. In addition, the revised abstract highlights the discussion of the methods proposed in this paper, providing a clearer introduction to the innovative points of this article. The revised abstract better meets the requirements of a professional academic paper.
The revised abstract can also be seen in the “Abstract” on page 1 of the revised manuscript and has been marked in blue.
(2) The discussion on the alignment technology of inertial navigation systems is too broad. It would be more effective if the paper focused on a specific aspect, such as process adaptability, multi-source information fusion, or rotational alignment during movement.
The author fully agrees with the reviewer's opinion that the discussion on inertial navigation system alignment technology is too broad, and there are indeed issues with lengthy content and unclear key points. Based on the review comments, we have made targeted modifications.
The various classification methods for initial alignment of inertial navigation systems are not the core focus of this paper. The original manuscript uses a large amount of space to elaborate on them, resulting in unclear key points and cumbersome content. This part of the content has been deleted in the revised manuscript. The core issue studied in this paper is in-motion initial alignment, therefore the introduction of static base, shaking base, and in-motion initial alignment is retained, which is the fundamental concept of the research work in this paper.
The discussion of research status and progress should focus on the core technology of multi-source information fusion. The original manuscript mistakenly mixed in the description of transmission alignment, and this part has also been deleted in the revised manuscript.
The focus of the revised manuscript is clearer, the logic is more reasonable, and the details are more appropriate. The paragraphs related to modifications have been marked in purple font in the revised manuscript. For details, please refer to the second and third paragraphs on page 2 of the revised manuscript.
(3) In the adaptive alignment flowchart, when parking time is greater than 1 minute but less than 5 minutes, flag = 0. Is it still necessary to conduct coarse alignment during movement in this case?
Due to the lack of clear and intuitive narration in the paper, it has caused confusion and trouble for the reviewers. The author deeply apologizes for this and provides a more detailed explanation here.
The original manuscript has already mentioned “the coarse alignment of the wobble base is considered to be completed if it is more than 1 min, corresponding to flag = 1”. Therefore, when the parking time is greater than 1 minute and flag = 1 instead of flag = 0. According to the technical solution proposed in this paper in the adaptive alignment flowchart, coarse alignment is no longer required at this time.
The position of the definition of flag in the revision has been changed and marked in orange. Please refer to the last paragraph on page 17 of the revision for details.
(4) At the beginning of page 20, it is noted that the odometer output is zero under parking conditions, which is consistent with the static zero-velocity assisted alignment model. Why is there a need for additional verification for parking? Does the odometer-assisted optimal estimation alignment dynamic model not accommodate static conditions?
The author deeply admires the rigorous academic attitude and profound professional knowledge of the reviewers, and pointed out the shortcomings of this manuscript to the point. The author provides the following explanations for the questions raised by the reviewers.
The odometer-assisted optimal estimation alignment dynamic model is aimed at the alignment scheme under the condition of carrier motion. The model considers position errors, and if the precise position is known, higher alignment accuracy will be achieved. However, in the static alignment model, position errors do not need to be considered, so there are subtle differences between the two models.
At the reminder of the reviewers, the author found a loophole in the description of the original manuscript. The precise position information cannot be obtained during the autonomous in-motion alignment process, and the impact of position errors on in-motion alignment and how to autonomously calculate the position during in-motion alignment are not within the scope of this paper. Therefore, this redundant discussion content has been deleted in the revision.
(5) The validity of using the alignment results from the combination of GNSS and inertial navigation systems during movement as a reference benchmark for autonomous alignment should be reconsidered. The GNSS-aided alignment accuracy is significantly influenced by carrier maneuverability and GNSS speed accuracy. Given that rotational inertial navigation is utilized, would it not be more appropriate to use the static rotational alignment results at the end of the in-motion alignment as a reference?
As the reviewer pointed out, the static rotation alignment result at the end of the motion alignment is the most accurate and obviously the most suitable reference benchmark for the alignment result. However, this benchmark can only be used at the final parking point, and it still takes some time to align after parking, which goes against the original intention of quick alignment and obtaining results by parking of in-motion initial alignment. The reasons for using the alignment results of GNSS and INS combination as the reference benchmark for autonomous alignment during movement are explained as follows.
On the one hand, INS has high accuracy in a short period of time, which can compensate for the drawbacks of GNSS's susceptibility to carrier maneuverability and environmental interference, while the high accuracy of GNSS can suppress the error divergence caused by INS's long-term use. In addition, the alignment scheme proposed in the manuscript is based on the static base alignment, and the alignment results have a certain guarantee. Therefore, the effectiveness of using the alignment results of GNSS and INS combination as the reference benchmark for autonomous alignment results during movement has a certain degree of reliability.
On the other hand, GNSS has certain accuracy advantages over multi-source information fusion schemes such as odometer and laser velocimeter. Therefore, the combination scheme of GNSS and INS can be used as a reference benchmark to verify the feasibility, effectiveness, and accuracy of the fully autonomous alignment scheme proposed in this manuscript.
Last but not least, Alignment during movement pays more attention to the changes and convergence trends of alignment results during the motion process, and achieves immediate acquisition of attitude alignment results at the moment of parking. Therefore, it is necessary to establish an attitude reference during the motion process. It is obviously very difficult to establish a benchmark for obtaining an absolutely accurate attitude, and using a combination alignment results of high-precision GNSS and highly reliable INS is a feasible method.
In summary, the static rotation alignment results at the end of motion alignment theoretically have higher accuracy alignment results, but are not suitable as attitude references for in-motion initial alignment. Therefore, we still prefer to use the combined alignment results of GNSS and INS as attitude references.
(6) There are missing units for the numerical values presented in Table 3, which should be addressed for clarity.
After being reminded by the reviewing experts, the author found that Table 3 did indeed omit the units of numerical values. The revised manuscript has improved this low-level error by adding the (′) for alignment results, which is highlighted in green in Table 3. For details, please refer to Table 3 on page 21 of the revised manuscript.

Reviewer 2 Report
Comments and Suggestions for Authors
The work concerns the initialization of the navigation system of a vehicle that is equipped with a redundant sensor system of a satellite navigation receiver, an odometer, and a Laser Doppler velocimeter. Initially, model equations for the motion of the vehicle are formulated in different coordinate systems. These model equations form the core of a model based state estimator. The Federal Kalman Filter approach is chosen, where one Kalman filter is set up for each sensor, and the estimates of the filters are fused to one joint estimate. The approach is tested in experiments with a vehicle.
The approach is not entirely new, but quite interesting and publishable. The paper should be revised in the following manner:
* The paper is rather long. Especially, Secion 2.2 presenting the equations of motion for the vehicle seems too detailed to me. I recommend shortening the section by refering to previous work and other literature.
* eq. (17): The symbols R_N and h are not explained.
* eq. (46) and (47): indices seem to be missing for P and Q and the right-hand sides of the equations.
* eq. (46) to (48): please explain the meaning of index m in a bite more detail.
* Figure 5: The two photographs of the vehicle seem identical to me. If not, please point out the difference.
* Figure 6: The vertical axis should be labeled "latitude".
* Table 2: The heading of the right-hand columns should read "*with* information multiplexing"
Author Response
The authors are very grateful to the reviewers for recognizing the research work in this letter. We have carefully considered and agreed with the issues raised by the reviewer and have made targeted revisions based on the review comments, as detailed below.
(1) The paper is rather long. Especially, Section 2.2 presenting the equations of motion for the vehicle seems too detailed to me. I recommend shortening the section by referring to previous work and other literature.
The reviewer pointed out that some of the content in this manuscript is described in a cumbersome manner, and the author strongly agrees with this. After checking the entire paper, the author found that other sections of the paper were also too broad and lengthy, resulting in insufficient emphasis on the key points of the paper. This issue was specifically addressed in the revised manuscript. On the one hand, some of the work is supplemented by references, and on the other hand, the core content of the alignment method for multi-source information fusion is discussed in detail, with a reduction of non-key content.
The rewritten and improved content has been marked in purple font in the revised manuscript, please refer to page 4, page 6, and page 8 of the revised manuscript for details.
(2) Eq. (17): The symbols RN and h are not explained.
The author deeply apologizes for the stupid error of lacking specific explanations for formula symbols in the manuscript, and has made supplements and modifications in the revision. The RM in the Eq. (17) represents the radius of curvature of the meridian circle, RN represents the curvature radius of the prime vertical, and the h is the local height.
In the process of revising the paper, some familiar and well-known formulas and symbols in the industry were provided through references. Equation (17) in the original manuscript has been deleted, so this part will no longer appear in the revised manuscript.
(3) Eq. (46) and (47): indices seem to be missing for P and Q and the right-hand sides of the equations.
At the reminder of the reviewers, the author found that the symbols used in Eq. (46) and (47) were indeed missing indices, which caused ambiguity and misunderstanding in the understanding of the Equations. Therefore, the targeted modifications were made in the revised manuscript. The equation labels have been revised to (33) - (34) and marked in green, and the modified equations are as follows.
(33)
(34)
Where n is the number of sub-filters, i represents the ith sub-filter, and m is the main filter.
(4) Eq. (46) to (48): please explain the meaning of index m in a bite more detail.
After inspection, it was found that the manuscript did lack an explanation of the meaning of index m. The revised manuscript made clear additions and marked them in green. Please refer to the green font between formulas (34) and (35) on page 11 of the revision for details.
(5) Figure 5: The two photographs of the vehicle seem identical to me. If not, please point out the difference.
At the reminder of the reviewers, the author found that there was indeed an error in the use of experimental equipment photographs during the writing process of the manuscript, mistakenly using the same photograph twice. The second photograph has been replaced in the revised manuscript, mainly to reflect the inertial navigation system equipment used in this experiment. Please refer to Figure 5 on page 18 of the revised manuscript for detailed information. The title of Figure 5 has been highlighted in yellow.
6) Figure 6: The vertical axis should be labeled "latitude".
As the reviewer pointed out, the vertical axis of roadmap should be “Latitude”, and the author mistakenly wrote “Longitude”, and we deeply apologize for this low-level error, which has been corrected in the revised manuscript. Please refer to Figure 6 on page 18 of the revised manuscript for details, and its title is also highlighted in yellow.
(7) Table 2: The heading of the right-hand columns should read "*with* information multiplexing".
According to the suggestions of the reviewers, the heading of the Table 2 have been revised to “ ‘with’ information multiplexing”, which have been highlighted in blue. Please refer to the Table 2 on page 19 of revision.

Round 2
Reviewer 1 Report
Comments and Suggestions for Authors
The two titles in the first row of Table 2 are identical; should they be modified? The title in the second row of the table indicates that the alignment results without information multiplexing remain in an inertial navigation state. Does this column's alignment result is obtained without any assistance of external sensors? If we are to compare methods with information multiplexing and without information multiplexing , shouldn’t the comparison be made under the same auxiliary information conditions?
After modifying or explaining the above issues, the manuscript can be accepted.
Author Response
(1) The two titles in the first row of Table 2 are identical; should they be modified? The title in the second row of the table indicates that the alignment results without information multiplexing remain in an inertial navigation state. Does this column's alignment result is obtained without any assistance of external sensors? If we are to compare methods with information multiplexing and without information multiplexing , shouldn’t the comparison be made under the same auxiliary information conditions?
Under the guidance of the reviewer, the authors did find a writing error in the title of the first row of Table 2. The revised manuscript changed the second title to "with information multiplexing" and highlighted it in yellow.
The reviewing experts have doubts about the experimental conditions for the results in Table 2. The alignment result without information multiplexing in the inertial frame was carried out with the assistance of LDV. After the author's inspection, this crucial statement was indeed overlooked in the original manuscript. OD or LDV can be used as external sensors for full inertial frame alignment. Compared to OD assistance, the LDV assisted information multiplexing and alignment effect of the experimental data is better. Therefore, in order to better compare and verify the effect of information multiplexing, the LDV displacement increment assisted full inertial frame is selected as the comparison for this experiment.
The revised manuscript not only supplements the description of this key experimental condition, but also improves the legend of Figure 8 and the title of Table 2, as indicated in red font on page 19 of the revised manuscript.

Reviewer 2 Report
Comments and Suggestions for Authors
I thank the authors for responding to my questions. I do not have further suggestions for improvement.
Author Response
I thank the authors for responding to my questions. I do not have further suggestions for improvement.
The authors are very grateful to the reviewers for their recognition of the research work, and your review comments have played a very important role in improving the professionalism and content quality of this paper.